# A common East-Asian *ALDH2* mutation causes metabolic disorders and the therapeutic effect of ALDH2 activators

Yi-Cheng Chang [1,2,3,10], Hsiao-Lin Lee[2,10], Wenjin Yang[4], Meng-Lun Hsieh[1,2], Cai-Cin Liu[1], Tung-Yuan Lee[1], Jing-Yong Huang[1,2], Jiun-Yi Nong[1,2], Fu-An Li [3], Hsiao-Li Chuang[5], Zhi-Zhong Ding[1], Wei-Lun Su[1], Li-Yun Chueh [1], Yi-Ting Tsai[6], Che-Hong Chen[7], Daria Mochly-Rosen [7] ✉ & Lee-Ming Chuang [2,8,9] ✉

Obesity and type 2 diabetes have reached pandemic proportion. ALDH2 (acetaldehyde dehydrogenase 2, mitochondrial) is the key metabolizing enzyme of acetaldehyde and other toxic aldehydes, such as 4-hydroxynonenal. A missense Glu504Lys mutation of the *ALDH2* gene is prevalent in 560 million East Asians, resulting in reduced ALDH2 enzymatic activity. We find that male *Aldh2* knock-in mice mimicking human Glu504Lys mutation were prone to develop diet-induced obesity, glucose intolerance, insulin resistance, and fatty liver due to reduced adaptive thermogenesis and energy expenditure. We find reduced activity of ALDH2 of the brown adipose tissue from the male *Aldh2* homozygous knock-in mice. Proteomic analyses of the brown adipose tissue from the male *Aldh2* knock-in mice identifies increased 4-hydroxynonenal-adducted proteins involved in mitochondrial fatty acid oxidation and electron transport chain, leading to markedly decreased fatty acid oxidation rate and mitochondrial respiration of brown adipose tissue, which is essential for adaptive thermogenesis and energy expenditure. **AD-9308** is a water-soluble, potent, and highly selective ALDH2 activator. **AD-9308** treatment ameliorates diet-induced obesity and fatty liver, and improves glucose homeostasis in both male *Aldh2* wild-type and knock-in mice. Our data highlight the therapeutic potential of reducing toxic aldehyde levels by activating ALDH2 for metabolic diseases.

Obesity and type 2 diabetes have reached pandemic levels. The 2019 International Diabetes Federation reported that there were 463 million people living with diabetes worldwide. This number is projected to increase rapidly to 578 million in 2030[1]. East Asia and South Asia, the most populous regions in the world, are expected to be the major contributors of new cases[1]. This pandemic is primarily caused by a high-calorie diet enriched in fat and sugar, sedentary lifestyle, and their interaction with genetic factors[1,2].

Approximately 36% of East Asians (560 million) or nearly 8% of the global population, carry an inactivating Glu504Lys missense mutation

[1]Graduate Institute of Medical Genomics and Proteomics, National Taiwan University, Taipei, Taiwan. [2]Department of Internal Medicine, National Taiwan University Hospital, Taipei, Taiwan. [3]Institute of Biomedical Sciences, Academia Sinica, Taipei, Taiwan. [4]Foresee Pharmaceuticals, Co.Ltd, Taipei, Taiwan. [5]National Laboratory Animal Center, Taipei, Taiwan. [6]Laboratory Animal Center, College of Medicine, National Taiwan University, Taipei, Taiwan. [7]Department of Chemical and Systems Biology, Stanford University School of Medicine, Stanford, CA, USA. [8]Graduate Institute of Molecular Medicine, National Taiwan University, Taipei, Taiwan. [9]Graduate Institute of Clinical Medicine, National Taiwan University, Taipei, Taiwan. [10]These authors contributed equally: Yi-Cheng Chang, Hsiao-Lin Lee. ✉e-mail: mochly@stanford.edu; leeming@ntu.edu.tw

of the *ALDH2* gene[3]. This mutation results in a reduction of the ALDH2 enzymatic activity by 60–80% in heterozygous carriers and ~90% in homozygous carriers[4]. People carrying this mutation exhibit sensitivity to alcohol, with presentations ranging from facial flushing, headache, and tachycardia due to a rapid increase in circulating acetaldehyde concentrations[5,6]. Epidemiological studies suggest a correlation between this inactivating mutation and several diseases including oral cancer, esophageal cancers, and blood and solid tumors associated with Fanconi anemia[7,8]. Importantly, several large-scale meta-analyses of genome-wide association studies revealed that this inactivating mutation is strongly associated with type 2 diabetes[9], body mass index[10], and serum lipids [11]in East Asians. A validation study further confirmed a close association of the *ALDH2* mutant variant with visceral fat distribution in 2958 Chinese subjects[12]. In addition, carriers of the inactivating *ALDH2* variant had 2-3 times increased risk of non-alcoholic fatty liver disease among Japanese subjects[13].

The *ALDH2* gene encodes mitochondrial aldehyde dehydrogenase 2 (ALDH2), a major acetaldehyde-metabolizing enzyme responsible for ~95% acetaldehyde metabolism due to its low Km for acetaldehyde[14]. Acetaldehyde is produced by alcohol dehydrogenase (ADH) following alcohol ingestion. Acetaldehyde can also be generated endogenously from the intermediate metabolisms or by gut microbial flora[15]. Air pollution, thermal degradation of plastics, and spoiled food are also sources of toxic aldehyde[16–18]. Acetaldehyde is listed as group I carcinogen by the International Agency for Research on Cancer[19]. In addition to acetaldehyde, ALDH2 also metabolizes various bioactive toxic aldehydes, including acrolein, malondialdehyde and 4-hydroxynonenal (4-HNE). Among these bioactive aldehydes, the most intensively studied one has been 4-HNE, a lipid peroxidation product that forms covalent adduct to macromolecules such as protein, lipid and DNA, causing cellular damage[20–24].

With the advent of high-throughput screening, a small molecule, **Alda-1**, was identified as an activator for both the wild-type ALDH2 and mutant ALDH2 enzymes[25]. Based on X-ray crystallography and enzyme kinetics, the binding of **Alda-1** partially blocks the exit tunnel of substrates, thus increasing the likelihood of productive encounters between reaction intermediate and the catalytic site [26]. Administration of **Alda-1** has been effective to treat myocardial infarction, aortic aneurysm, atrial fibrillation, Alzheimer's disease and nociceptive pain by enhancing the clearance of 4-HNE in rodents[27–30]. The discovery of ALDH2 activators offers an opportunity to test the therapeutic potential of reducing toxic aldehydes for treating a variety of diseases.

**AD-9308** is a highly water soluble and orally bioavailable prodrug of a potent and highly selective ALDH2 activator, **AD-5591**, a new generation ALDH2 activator that has the improved biological activities and pharmacological properties compared to **Alda-1**[31].

In this study, we used *Aldh2* homozygous knock-in (KI) and heterozygous knock-in (HE) mice, which mimic the East Asian-specific Glu504Lys mutation, to evaluate the effect of this mutation on diet-induced obesity, glucose homeostasis, fatty liver, and serum lipids, and tested the therapeutic effect of a feasible ALDH2 activator **AD-5591** by dosing its prodrug **AD-9308** for treating metabolic disorders.

## Results

### *Aldh2* KI mice carrying the East Asian-specific Glu504Lys mutation were prone to develop diet-induced obesity and fatty liver

To assess the effect of the East Asian-specific *ALDH2* Glu504Lys mutation on diet-induced obesity and related metabolic traits, *Aldh2* KI, HE, and WT mice were placed on high-fat high-sucrose diet (HFHSD) for 24 weeks since the age of 4 weeks. At the end of study, the body weights of *Aldh2* KI, HE and WT mice were $41.66 \pm 1.40$, $41.65 \pm 1.13$, and $36.95 \pm 0.96$ g respectively (Fig. 1a, *P*-for-trend = 0.0096). However, no difference of body weight gain was observed between *Aldh2* KI and WT mice fed a chow diet. *Aldh2* KI, HE, and WT mice fed on HFHSD also had

more white fat, including perigonadal fat ($2.76 \pm 0.27$ vs. $2.41 \pm 0.16$ vs. $1.63 \pm 0.13$ g, *P*-for-trend < 0.001), inguinal fat ($1.69 \pm 0.16$ vs. $1.55 \pm 0.12$ vs. $1.24 \pm 0.11$ g, *P*-for-trend = 0.011), mesenteric fat ($0.85 \pm 0.09$ vs. $0.87 \pm 0.05$ vs. $0.32 \pm 0.05$ g, *P*-for-trend = 0.018), and a trend of increased liver weight ($1.84 \pm 0.07$ vs $1.70 \pm 0.13$ g vs. $1.61 \pm 0.07$ g, *P*-for-trend = 0.069), but less brown adipose tissue (BAT) weight ($0.12 \pm 0.009$ vs. $0.17 \pm 0.0.02$ vs. $0.17 \pm 0.01$ g, *P*-for-trend = 0.006) (Fig. 1b) at the age of 24 weeks. No such differences were found between *Aldh2* KI and WT mice on chow diet. Body composition analysis revealed significantly increased fat mass of *Aldh2* KI and HE compared with the WT mice fed on HFHSD ($12.40 \pm 1.00$, $10.04 \pm 1.09$, $8.41 \pm 0.78$ g, *P*-for-trend = 0.0058) (Fig. 1c) at the age of 24 weeks. There was no difference of fat mass between *Aldh2* KI and WT mice on a chow diet. Figure 1d showed the representative gross appearance of mice, BAT, perigonadal fat, and liver at the age of 24 weeks. There was dose-dependent increase in adipocyte size ($1039.2 \pm 54.6$, $1247.9 \pm 91$, $1433.6 \pm 91.22$ $\mu m^2$, *P*-for-trend = 0.0013) among three genotype although there was no difference in the number of adipocytes (Fig. 1e, f), indicating hypertrophy rather than hyperplasia of white adipose tissue. *Aldh2* KI and HE mice also had significantly higher hepatic triglycerides contents and more severe hepatic steatosis than the WT mice on HFHSD ($0.32 \pm 0.05$ vs, $0.21 \pm 0.03$ vs. $0.19 \pm 0.03$ mg/mg liver tissue, *P*-for-trend = 0.034) (Fig. 1g, j) at the age of 24 weeks. The skeletal muscle triglycerides content is also higher in *Aldh2* KI and HE mice compared to WT mice ($0.18 \pm 0.01$ vs, $0.15 \pm 0.02$ vs. $0.10 \pm 0.01$ mg/mg muscle tissue, *P*-for-trend = 0.0051) (Fig. 1h). Hematoxylin and eosin (H&E) stain of perigonadal fat showed hypertrophic adipocytes with more crown-like necrosis in *Aldh2* KI and HE mice compared with controls at the age of 24 weeks (Fig. 1i). These results suggest that the East Asian-specific ALDH2Glu504Lys mutation promoted HFHSD-induced obesity and fatty liver in mice.

### *Aldh2* KI mice had reduced energy expenditure and impaired adaptive thermogenesis

Energy expenditure measured by indirect calorimetry showed that *Aldh2* KI and HE mice had significantly lower energy expenditure, especially in active (dark) phase at the age of 8–10 weeks (Fig. 2a) than WT mice. In addition, indirect calorimetry also revealed *Aldh2* KI and HE mice have higher respiratory exchange rate (RER) compared with WT mice (Supplementary Fig. 1), suggesting reduced fatty acid utilization. There was no difference in accumulative food intake over three weeks among three genotypes (Fig. 2b). These data indicated *Aldh2* KI and HE mice were prone to diet-induced obesity due to reduced energy expenditure but not intake.

Energy expenditure is composed of resting metabolic rate, physical activity, and adaptive thermogenesis including cold-induced and diet-induced thermogenesis. Indirect calorimetry showed lower energy expenditure of *Aldh2* KI and HE mice compared with WT mice. The difference is substantial during active phase (night time). Nevertheless, there is still a small but statistically difference in resting metabolic rate during inactive phase (day time) (Fig. 2a). There was no difference of accumulative food intake measured for three weeks among three genotypes (Fig. 2b). We also measured energy expenditure by monitoring physical activities of the mice at the age of 8–10 weeks. Daily physical activity, including wheel rotations (Fig. 2c) and travel distances measured by HomeCage monitoring system (Fig. 2d) were similar among the three groups. HomeCage monitoring systems did not detect significant differences in various mouse behaviors including awakening, drinking, feeding, grooming, hanging, resting, twisting, walking, and rearing up (Fig. 2e).

Consequently, we measured adaptive thermogenesis, including cold-induced, diet-induced, and isoproterenol-induced thermogenesis at the age of 8–10 weeks. For diet-induced thermogenesis, *Aldh2* KI and HE mice exhibited significantly lower rectal temperature after HFHSD feeding than WT mice at the age of 20–24 weeks, indicating impaired diet-induced thermogenesis (Fig. 2f). Compatible with rectal

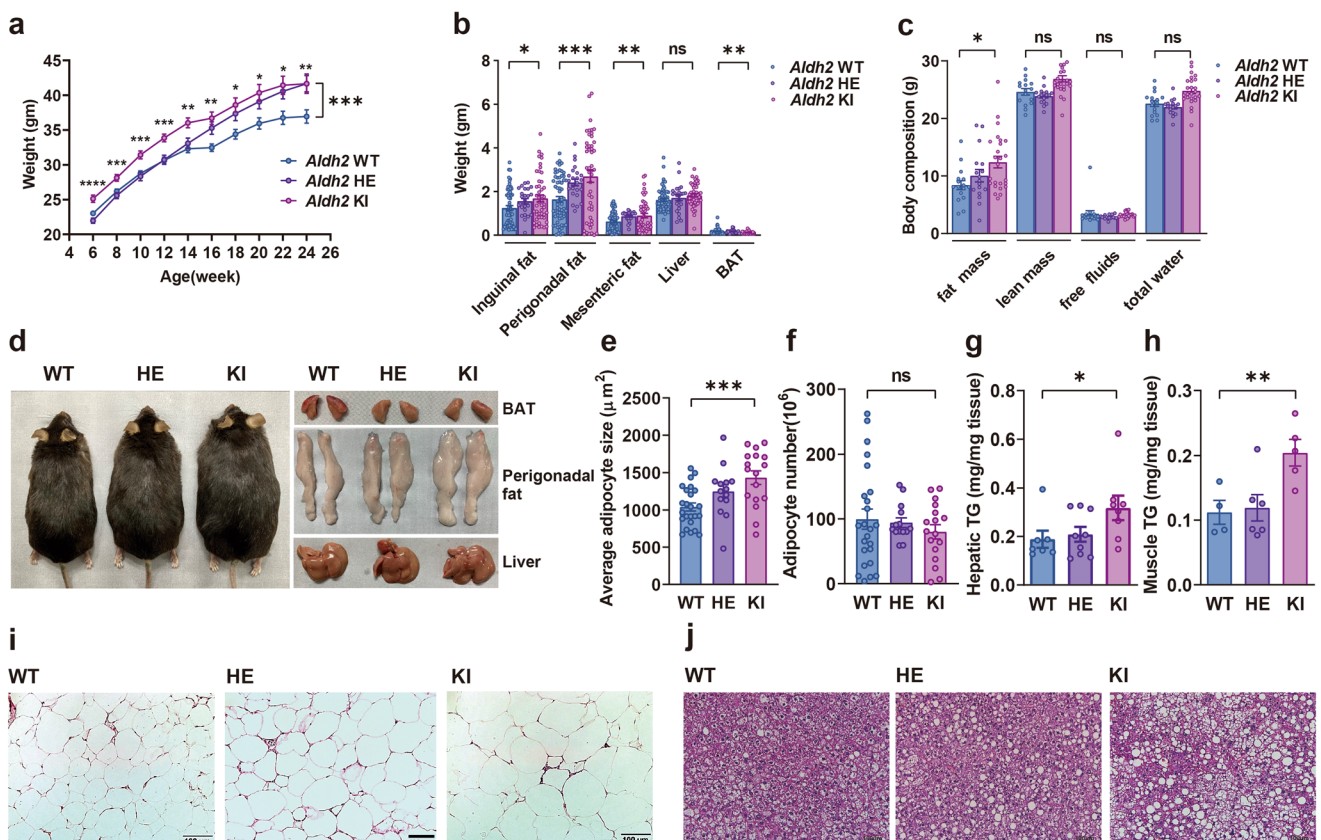

**Fig. 1 | *Aldh2* (acetaldehyde dehydrogenase 2) knock-in (KI) mice mimicking human Glu504Lys mutation were prone to develop diet-induced obesity, glucose intolerance, insulin resistance, and fatty liver.** Body weight of *Aldh2*, homozygous knock-in (KI) and heterozygous KI (HE), and wild-type (WT) mice on **a** high-fat high sucrose diet (HFHSD) ($n = 24:36: 21$; repeated measure analysis of variance [ANOVA] $P = 0.0002$) and **b** weights of inguinal, perigonadal, and mesenteric fat, the liver, and brown adipose tissue (BAT) of the *Aldh2* WT and KI mice ($n = 63:24:48$; $P$-for-trend = 0.011, <0.001, 0.0065, 0.053, and 0.0069 respectively). **c** Body composition of *Aldh2* WT and KI mice ($n = 16:17:22$ in triplicates; $P$-for-trend = 0.02, 0.09, 0.77, 0.10 and respectively). **d** Gross appearance if mice, BAT, perigonadal fat and liver. **e** Average adipocyte size ($P$-for-trend = 0.0003) and **f** adipocyte number in the perigonadal fat pad of the *Aldh2*

homozygous KI, HE, and WT mice ($n = 24:14:17$; $P$-for-trend = 0.030). **g** Hepatic triglyceride content of the *Aldh2* homozygous KI, HE, and WT mice ($n = 7:9:8$ in duplicates; $P$-for-trend = 0.034). **h** Muscle triglyceride content in the *Aldh2* homozygous KI and HE, and WT mice on HFHSD ($n = 7:8:8$; $P$-for-trend = 0.0086). **i** Representative H&E (hematoxylin and eosin) stain of the perigonadal fat in *Aldh2* KI, HE, and WT mice. **j** Representative H&E staining of livers from the *Aldh2* homozygous KI and HE and WT mice. The scale bar of is 100 μm. The scale bar of is 100 μm. Figures (**a**–**c**, **e**–**h**) were analyzed tests for linear trends. Figure (**a**) was further analyzed using repeated measures ANOVA. All data are presented as mean and standard error (S.E.M.). The *n* values represent biological repeats and the number of technical repeats were expressed as duplicates or triplicates. The asterisks indicate two-sided *$P < 0.05$, **$P < 0.01$, ***$P < 0.001$.

temperature, indirect calorimetry also revealed lower energy expenditure after diet intake of *Aldh2* KI and HE mice compared with WT mice (Supplementary Fig. 2).

For cold tolerance test, *Aldh2* KI and HE mice had lower rectal temperature during 18-h prolonged cold exposure at 4 °C (Fig. 2g) compared with WT mice at the age of 20–24 weeks. However, there was no difference of rectal temperature in 4-h acute cold tolerance test among three genotypes (Supplementary Fig. 3a). Consistently, there was also little difference of energy expenditure in the acute cold tolerance test among three genotypes (Supplementary Fig. 3b). Furthermore, *Aldh2* KI and HE mice also have significantly lower isoproterenol-induced energy expenditure than WT mice (Fig. 2h). However, there was no significance in serum norepinephrine among three genotypes (Supplementary Fig. 4).

These data suggest that *Aldh2* KI mice may develop obesity due to reduced energy expenditure resulting from impaired adaptive thermogenesis.

### *Aldh2* KI mice displayed reduced insulin sensitivity and impaired glucose tolerance

Since the *ALDH2* Glu504Lys mutation is reported to be associated to type 2 diabetes in genome-wide association studies in East-Asians[9], we

further examined glucose homeostasis in mice. On HFHSD, insulin tolerance test (ITT)showed significantly higher blood glucose levels of *Aldh2* KI and HE mice than WT mice, indicating increased insulin resistance (Fig. 2i). Intraperitoneal glucose tolerance (i.p. GTT) showed significantly higher blood glucose levels of *Aldh2* KI and HE mice than WT mice at the age of 24 weeks (Fig. 2j). Oral glucose tolerance test (OGTT) also showed similar findings (Fig. 2k). *Aldh2* KI mice displayed a compensatory increase in insulin secretion after oral glucose load (Fig. 2l). No differences in glucose homeostasis measured by glucose and insulin tolerance tests were found between *Aldh2* KI and WT mice fed a chow diet.

There were no difference in serum total cholesterols(Supplementary Fig. 5a) and triglycerides levels (Supplementary Fig. 5b) among three genotypes at the age of 24 weeks. However, *Aldh2* KI and HE mice showed dose-dependently increase in serum leptin levels ($11.13 \pm 2.13$ vs. $7.21 \pm 0.97$vs.$3.24 \pm 0.39$ mg/ml, $P$-for-trend = 0.0002, Fig. 2m), increase in free fatty acid levels ($0.92 \pm 0.040$ vs. $1.08 \pm 0.05$ vs. $0.83 \pm 0.05$ mEq/L $P$-for-trend = 0.012, Fig. 2n), decrease in serum adiponectin levels ($7832.7 \pm 106.4$ vs. $8539.8 \pm 312.4$ vs. $9115.6 \pm 356.9$ ng/ml, $P$-for-trend = 0.0032 Fig. 2o), and increase ethanol levels ($0.63 \pm 0.09$ vs. $0.40 \pm 0.06$ vs. $0.34 \pm 0.07$ mEq/L $P$-for-trend = 0.0008, Fig. 2p) compared with WT mice.

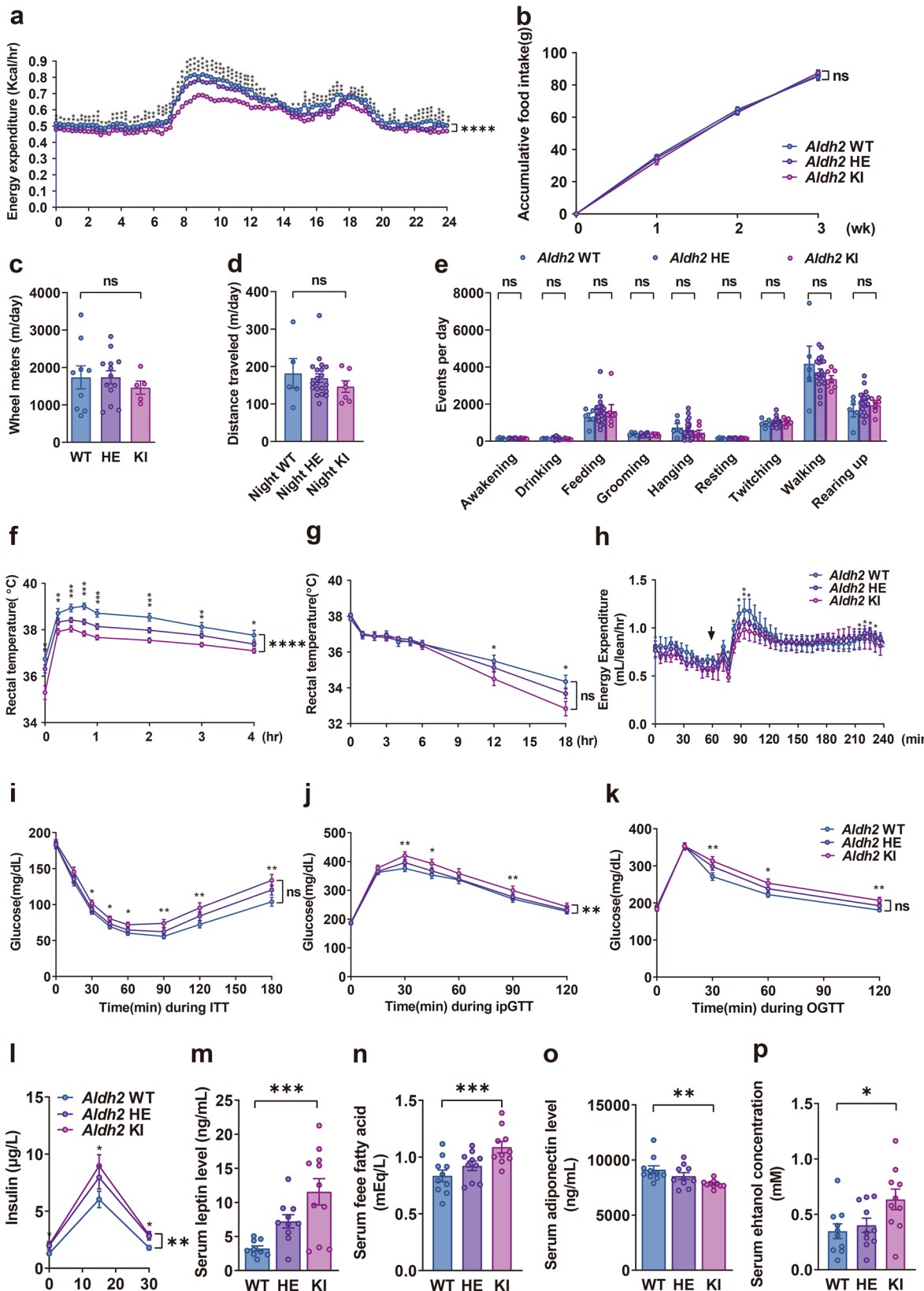

**Aldh2 KI mice had reduced mitochondrial fatty acid oxidation rate and lower respiratory transport chain activity in brown adipose tissue due to 4-HNE adduction**

In view of the reduced adaptive thermogenesis observed in the *Aldh2* KI mice, we compared the expression levels of protein involved in

thermogenesis including *Ucp1*, *Pgc1α*, *Prdm16*, *Cidea*, and *Dio2* between their BATs using immunoblots and real-time quantitative PCR (RT-qPCR) at the age of 24 weeks. There was no difference of these proteins between BAT isolated from *Aldh2* KI and WT mice (Fig. 3a, b, Supplementary Fig. 6). Furthermore, there was no difference in gene

**Fig. 2 | *Aldh2* (acetaldehyde dehydrogenase 2) knock-in mice mimicking human Glu504Lys mutation have reduced energy expenditure and adaptive thermogenesis. a** Energy expenditure ($n = 19:27:13$; repeated measures analysis of variance [ANOVA] $P < 0.0001$), **b** accumulative food intake for three weeks ($n = 8:24:10$; repeated measures ANOVA $P = 0.70$), **c** wheel rotations ($n = 9:13:5$; P-for-trend = 0.55), **d** distance traveled ($n = 5:20:7$ P-for-trend = 0.27), and **e** behaviors monitored by HomeCage systems ($n = 5:20:7$; P-for-trend = 0.88, 0.54, 0.48, 0.12, 0.31, 0.92, 0.65, 0.19, and 0.52 respectively) of the *Aldh2* homozygous knock-in (KI), heterozygous KI (HE), and wild-type (WT) mice. **f** Diet-induced thermogenesis measured by rectal temperature after high-fat high-sucrose feeding (repeated measures ANOVA $P < 0.0001$) and **g** cold-induced thermogenesis measured by rectal temperature of mice in 4 °C condition ($n = 9:24:10$; repeated measures ANOVA $P = 0.29$). **h** Isoproterenol-induced energy expenditure ($n = 4:8:4$) of *Aldh2* homozygous KI, HE, and WT mice. The arrow head indicates the injection time (repeated measures ANOVA $P = 0.15$). **i** Glycemic levels during the insulin sensitivity test (repeated measures ANOVA $P = 0.055$) and **j** intraperitoneal glucose tolerance tests (repeated measures ANOVA $P = 0.0049$), and **k** glycemic levels during the oral glucose tolerance test of *Aldh2* KI, HE, and WT mice ($n = 49:24:42$; repeated measures ANOVA $P = 0.053$). **l** Insulin levels following oral glucose load ($n = 35:14:44$; repeated measures ANOVA $P = 0.0048$ in duplicates). Fasting serum **m** leptin ($n = 9:10:11$; P-for-trend = 0.0002 in duplicates), **n** free fatty acid, and ($n = 10:10:10$; P-for-trend = 0.0008 in duplicates), **o** adiponectin ($n = 10:10:10$; P-for-trend = 0.0008 in duplicates), and **p** ethanol ($n = 10:10:10$; P-for-trend = 0.0032 in duplicates) concentration of *Aldh2* homozygous KI, HE, and WT mice on high-fat high-sucrose diet. Figure (**a**) was analyzed using the generalized linear model. Figures (**a**, **b**, **f**–**l**) were further analyzed using repeated measures ANOVA. Figures (**b**–**p**) were analyzed using tests for linear trend. All data are presented as mean and standard error (S.E.M.). The n values represent biological repeats and the number of technical repeats were expressed as duplicates or triplicates. The asterisks indicate two-sided *$P < 0.05$, **$P < 0.01$, ***$P < 0.001$, ****$P < 0.001$.

---

expression of *Ucp1* in white fat including inguinal and perigonadal fat between the *Aldh2* KI and WT mice measured by RT-qPCR (Supplementary Fig. 7) at the age of 24 weeks. In addition, we did not observe difference in primary brown adipocyte differentiation isolated from *Aldh2* KI and WT mice at the age of 4–6 weeks (Supplementary Fig. 8).

ALDH2 enzymatic activity of recombinant *Aldh2* KI protein is drastically reduced compared with WT proteins (Fig. 3c). Similarly, the ex vivo ALDH2 enzymatic activity of BAT isolated from *Aldh2* KI mice is also lower than those from WT mice at the age of 24 weeks (Fig. 3d). This is accompanied with higher ex vivo 4-HNE level of BAT isolated form *Aldh2* KI mice compared with WT mice (Fig. 3e). We next examined the expression of mitochondrial respiratory complex I to V component in BAT from mice at the age of 24 weeks. These protein components are essential for the maintenance of the proton gradient in mitochondrial electron transport chain (ETC) and are required for UCP1-mediated thermogenesis. There was no difference in mitochondrial respiratory complex I to V protein components expression in the BAT of *Aldh2* KI and WT mice (Supplementary Fig. 9). We also observed no difference in microscopic morphology of BAT, the major thermogenic organ at the age of 24 weeks (Supplementary Fig. 10).

Since the ETC protein expression studied showed no difference between *Aldh2* KI and WT mice, it implied that the decreased energy expenditure and thermogenesis were likely due to impaired protein functions. We found increased protein carbonylation (Fig. 3f, h), 4-HNE-adducted mitochondrial proteins, and 4-HNE concentration (Fig. 3g, h) and decreased ALDH2 enzymatic activity in the BAT from the KI mice compared with WT mice at the age of 24 weeks. Using liquid-chromatography tandem mass spectrometry (LC MS/MS) analysis, we identified nineteen 4-HNE-adducted BAT mitochondrial proteins in *Aldh2* KI mice and nine 4-HNE adducted mitochondrial proteins in WT mice, with eight proteins which are present in both *Aldh2* KI and WT mice at the age of 24 weeks (Fig. 3i, j).

Three identified adducted sites were present in proteins involved in fatty acid oxidation (FAO) including 3-ketoacyl-CoA thiolase and propionyl-CoA carboxylase. Eleven sites were present in proteins involved in the maintenance of mitochondrial electron transport chain (ETC) including NADPH dehydrogenase (complex I), succinate dehydrogenase (complex II), cytochrome b-1 complex (complex III), ATP synthase, and glycerol-3-phoshphate dehydrogenase. Three adducted sites were present in the mitochondrial contact site and cristae organizing system (MICOS) complex, which are essential for the maintenance of inner mitochondrial membrane structure (Supplementary Table S2a). Hence, we further evaluated the capacity of mitochondrial FAO and ETC to determine the effect of 4-HNE modification of these proteins at the age of 24 weeks. The ex vivo FAO rate of the whole BAT tissue isolated from *Aldh2* KI mice was significantly decreased by 70.0% ($P < 0.001$) compared with those from WT mice, which could be rescued by addition of 0.1, 0.2 and 0.4 µCi ³H-palmitate dose-dependently (P-for-trend = 0.02)

(Fig. 3k). Consistent with the whole BAT tissue, the FAO rate of the cultured primary brown adipocytes isolated from the *Aldh2* KI mice at the age of 4–6 weeks was also significantly decreased by 39.3% ($P = 0.02$) compared with those from WT mice, which could be rescued by addition of 0.2 and 0.6 µCi ³H-palmitate dose-dependently (P-for-trend = 0.045) (Fig. 3l). However, there was no difference of FAO in skeletal muscle and liver between KI and WT mice (Supplementary Fig. 11). Furthermore, addition of 4-HNE decreased FAO rate of induced primary brown adipocytes isolated from WT mice in a dose-dependent manner; FAO rates were reduced by 36% ($P < 0.01$) and 60% at 5 and 10 µM 4-HNE, respectively ($P < 0.01$) (Fig. 3m). In line with the LC-MS/MS finding, we found a significant reduction in the enzymatic activity of mitochondrial respiratory complexes I, II, and III, but not complex IV, in *Aldh2* KI mice compared with WT mice at the age of 24 weeks (Fig. 3n). Oxygen consumption rate was also significantly decreased in induced primary brown adipocytes isolated from *Aldh2* KI mice compared with WT mice at the age of 4 weeks (Fig. 3o). Since FAO is the major energy source of thermogenesis and the mitochondrial ETC is essential for maintaining proton gradient required for thermogenesis, the increased protein carbonylation, especially 4-HNE-adduction may explain the impaired thermogenesis of *Aldh2* KI mice.

## ALDH2 activator AD-9308/AD-5591 activated wild-type and mutant human ALDH2 enzymatic activity

**AD-9308** is a valine ester prodrug of a potent and selective small molecule ALDH2 activator **AD-5591**. When administered in vivo, **AD-9308** is rapidly converted to **AD-5591** by esterase hydrolysis (Fig. 4a). In vitro, **AD-5591** treatment increases the catalytic activity of recombinant wild-type and mutant human ALDH2 (Fig. 4b). In vivo, **AD-9308** administration showed a favorable pharmacokinetic profile in mice when administered orally or intravenously with high bioavailability (Supplementary Table 3).

**Alda-1** allosterically activates ALDH2 mainly by partially blocking the substrate exit tunnel, thereby accelerating the substrate-enzyme collision without impeding the catalytic sites of Cys302 and Glu268;[26] **Alda-1** also inhibits substrate-induced ALDH2 inactivation by protecting Cys301 and Cys303 oxidation[25,30]. Using molecular docking, we found that the binding pocket for **AD-5591** (Fig. 4c) is close to that of **Alda-1** (Supplementary Fig. 12a). Similar to **Alda-1** (Supplementary Fig. 12b), **AD-5591** is bound within a hydrophobic collar by Tyr456, Val458, Lys127, Met124, Gln462, Gly460, Phe459, Val120, and Phe292 of human ALDH2 (Fig. 4d). This binding site leaves the catalytic sites Cys302 and Glu268 unimpeded (Fig. 4e, Supplementary Fig. 12c). Therefore, the mechanism by which **AD-5591** activates ALDH2 is very similar to **Alda-1**. Furthermore, **Alda-1** forms a single hydrogen bond with the Asp457 residue of ALDH2. Yet, in our modeling analyses, **AD-5591** forms one additional hydrogen bond with Ala461 (Fig. 4e), which may explain its higher affinity to ALDH2 than **Alda-1**.

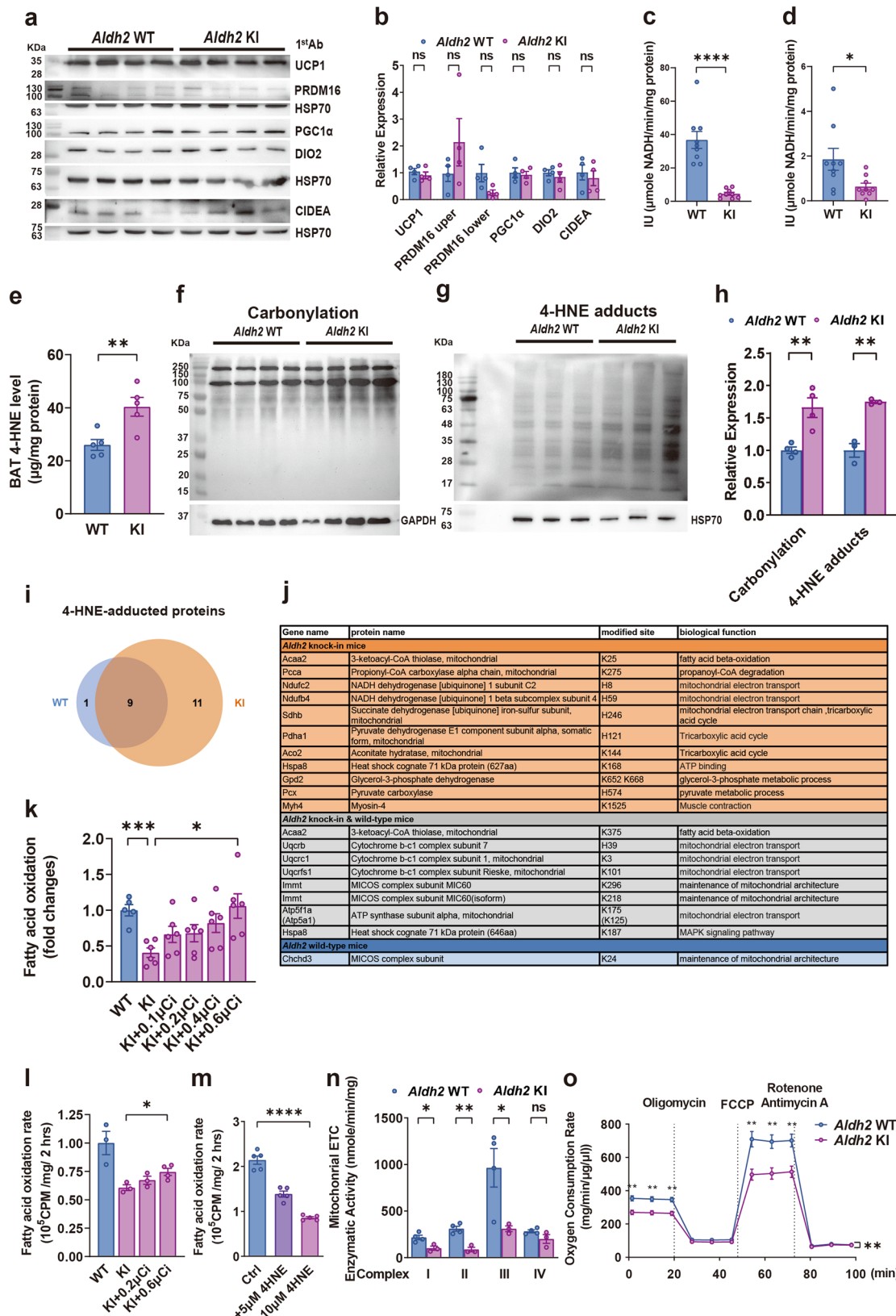

**ALDH2 activator AD-9308/AD-5591 lowered 4-HNE and attenuated diet-induced obesity, fatty liver, insulin resistance, and glucose intolerance in both WT and *Aldh2* KI mice**

In view of the marked reduction in Aldh2 expression in diet-induced obese mice (Supplementary Fig. 13), we next explored whether Aldh2 activation can rescue these obesity-associated phenotypes. From the

age of 10 weeks when HFHSD was started, *Aldh2* KI and WT mice were treated with vehicle, 20 mg/kg/day or 60 mg/kg/day of **AD-9308** by oral gavage for 20 weeks (Fig. 5a). As shown in Fig. 5b–e, **AD-9308** treatment effectively reduced the diet-induced weight gain in both *Aldh2* KI and WT mice. Weight of perigonadal fat, inguinal fat, omental fat decreased with **AD-9308** treatment dose-dependently (Fig. 5e–h)

**Fig. 3 | Aldh2 (acetaldehyde dehydrogenase 2) knock-in mice displayed impaired decreased fatty acid oxidation rate and mitochondrial respiration of brown adipose tissue due to increased 4-hydroxynonenal-adducted proteins.** **a** Immunoblots and densitometry histogram showing expression of *Ucp, Prdm16, Cidea, Dio2* levels (*n* = 4:4) and **b** densitometric histogram of brown adipose tissue (BAT) from *Aldh2* (wild-type) WT and (knock-in) KI mice (*P* = 0.42, 0.25, 0.07, 0.74, 0.50, and 0.61 respectively). **c** ALDH2 enzymatic activity of recombinant WT and KI ALDH2 protein (*n* = 6:6 in duplicates; *P* < 0.0001) and **d** BAT from *Aldh2* WT and KI mice (*n* = 9:8; *P* = 0.29). **e** 4-HNE (4-hydroxynonenal) concentration of BAT isolated from of the *Aldh2* WT and KI mice (*n* = 5:5; *P* = 0.008). **f** Protein carbonylation of BAT mitochondria by hydrazide biotin staining (*n* = 4:4) and **g** immunoblots and densitometry histogram of 4-HNE-adducted mitochondrial proteins in the BAT from the *Aldh2* WT and KI mice (*n* = 3:3). **h** densitometric histogram of **f** and; *P* = 0.006 and 0.0022 respectively) (**g**). **i, j** 4-HNE-adducted mitochondrial proteins and amino acid residues of the BAT from the *Aldh2* KI and WT mice, as identified by liquid-chromatography tandem mass spectrometry (LC-MS/MS) (*n* = 2:2). **k** Fatty acid oxidation (FAO) rate of the whole BAT tissue isolated from the *Aldh2* WT and KI mice rescued with 0.1. 0.2 and 0.4 μCi³H-palmitate respectively (*n* = 5:7:7:7:7;

*P* = 0.0003 and 0.0011). **l** FAO of primary brown adipocytes isolated from the *Aldh2* WT and KI mice rescued with 0.2 and 0.6 μCi ³H-palmitate respectively (*n* = 3:3:3:4; *P* = 0.016). **m** FAO rate of induced primary brown adipocytes isolated from the WT mice treated with 0, 5, and 10 μM 4-HNE (*n* = 3:3; *P* < 0.0001) rescued with plamitate. **n** Mitochondrial electron transfer chain (ETC) complex I-IV enzymatic activity isolated from the BAT of *Aldh2* WT and KI mice on HFHSD (*n* = 4:4; *P* = 0.019, 0.0018, 0.044, and 0.11 respectively). **o** Oxygen consumption rate (OCR) of induced primary brown adipocytes isolated from *Aldh2* WT and KI mice fed on high-fat high-sucrose diet. Cells were treated with oligomycin, FCCP (carbonyl cyanide p-trifluoro methoxyphenylhydrazone), and Rotenone/Antimycin A.OCR measurements were normalized to the protein concentration. (*n* = 3 per group; repeated measures analysis of variance (ANOVA) *P* = 0.0026). Figures (**k–m**) were analyzed using tests for linear trend. Figures (**b–e, h, k, n, o**) were analyzed using two-sample independent *t*-tests. Figure (**o**) was analyzed using repeated measures ANOVA. All data are presented as mean and standard error (S.E.M.). The n values represent biological repeats and the number of technical repeats are expressed as duplicates or triplicates. The asterisks indicate two-sided *P < 0.05, **P < 0.01, ***P < 0.001, ****P < 0.0001.

---

with reduced diet-induced adipocyte hypertrophy (Fig. 5i). AD-9308 treatment reduced the extent of hepatic steatosis (Fig. 5j, k, l) and the levels of hepatic triglycerides contents in both *Aldh2* KI and WT mice in a dose-dependent manner (Fig. 5m).

Fasting glucose levels were lowered by **AD-9308** treatment in both *Aldh2* WT and KI mice (Fig. 6a, b). **AD-9308** treatment effectively reduced insulin resistance in both WT (Fig. 6c) and KI (Fig. 6d) mice and significantly improved glucose tolerance of both WT (Fig. 6e) and KI (Fig. 6f) mice in a dose-dependent manner. **AD-9308** also decreased serum 4-HNE levels in both mice groups dose-dependently (Fig. 6g) at the age of 30 weeks. Figure 6h shows the summary diagram depicting how reducing 4-HNE by ALDH2 activator **AD-9308** ameliorates metabolic disturbances.

Pathological examination revealed no abnormalities in liver or kidney in both *Aldh2* WT and KI mice treated with **AD-9308** for 20 weeks (Supplementary Table 4). Serum alanine aminotransferase (ALT) and creatinine levels were also not different between groups after **AD-9308** treatment for 20 weeks (Supplementary Fig. 14).

Finally, to identify 4-HNE-adducted proteins modified by **AD-9308** treatment, we performed LC-MS/MS analyses of the BAT isolated from *Aldh2* WT and KI mice treated or not treated with **AD-9308** at the age of 30 weeks. As shown in Supplementary Fig. 15a and Supplementary Table S2b, treatment of **AD-9308** reduced the number of 4-NHE-adducted proteins in the BAT from *Aldh2* WT mice from fourteen to four. Among these ten proteins, eight proteins are involved in mitochondrial ETC and FAO including the Trifunctional enzyme subunit alpha, mitochondrial (Hadha), Succinate dehydrogenase [ubiquinone] flavoprotein subunit, mitochondrial (Sdha), Cytochrome b-c1 complex subunit 1, mitochondrialUqcrc1, Long-chain-fatty-acid--CoA ligase(Acsl1), Trifunctional enzyme subunit beta, mitochondrial(Hadhb), 3-ketoacyl-CoA thiolase, mitochondrial (Acaa2), Cytochrome b-c1 complex subunit 7 (Uqcrb), and Cytochrome c oxidase subunit 6C (Cox6c) being eliminated by AD9308 (Supplementary Fig. 15b).

Similarly, treatment of **AD-9308** reduced the number of 4-NHE-adducted proteins in the BAT from *Aldh2* KI mice from eighteen to four (Supplementary Fig. 16a). Among these fourteen proteins, twelve proteins are involved in mitochondrial ETC and FAO including the Trifunctional enzyme subunit alpha, mitochondrial (Hadha), Succinate dehydrogenase [ubiquinone] flavoprotein subunit, mitochondrial (Sdha), Cytochrome b-c1 complex subunit 1, mitochondrial (Uqcrc1), Cytochrome b-c1 complex subunit 2, mitochondrial (Uqcrc2), Trifunctional enzyme subunit beta, mitochondrial (Hadhb), Cytochrome c oxidase subunit NDUFA4 (Ndufa4), Mitochondrial carnitine/acylcarnitine carrier protein (Slc25a20), Cytochrome b-c1 complex subunit 7 (Uqcrb), NADH dehydrogenase [ubiquinone] 1 beta subcomplex

subunit 10 (Ndufb10), NADH dehydrogenase [ubiquinone] iron-sulfur protein 6, mitochondrial (Ndufs6), and Cytochrome c oxidase subunit 6C(Cox6c) being eliminated by AD-9308 (Supplementary Fig. 16b and Supplementary Table S2c). These results were consistent with the results comparing 4-HNE-adduced proteins of BAT between *Aldh2* WT and KI mice, showing ALDH2 modulates mitochondrial ETC and FAO function in BAT.

## Discussion

The prevalence of obesity and diabetes mellitus has surged in the past decades and is predicted to continue to rise, especially in East and South Asia[1]. This trend is largely caused by high-calorie diet enriched in fat and sugar, sedentary lifestyle, and their interaction with genetic predisposition[1,2]. Genome-wide association studies have confirmed many genetic loci associated type 2 diabetes and obesity. Specifically, several genetic loci are East Asian-specific. Genetic variants in or near the *CDKAL1, KLF9, GP2, ALDH2,* and *ITIH4* genes are associated with obesity[32,33] and genetic variants in or near the *GDAP1, PTF1A, SIX3, ALDH2,* and *PAX4* genes[9,34] are specifically associated with type 2 diabetes in East Asians. Among them, the *ALDH2* Glu504Lys mutation that affects 560 million East Asians or nearly 8% of global population is associated with body mass index, type 2 diabetes, and serum lipids in large meta-analyses of genome-wide association studies.

We demonstrated that *Aldh2* KI mice carrying the East Asian-specific Glu504Lys mutation were more prone to develop diet-induced obesity, fatty liver, insulin resistance and glucose intolerance than WT mice Importantly, the ALDH2 activator **AD-9308** increased both the catalytic activity of WT and mutant enzyme, reduced serum 4-HNE levels, and effectively alleviated diet-induced obesity, fatty liver, insulin resistance, and glucose intolerance in both *Aldh2* KI and WT mice in a dose-dependent manner.

BAT is a highly specialized organ enriched in Ucp1 for adaptive thermogenesis. Although *Aldh2* KI mice exhibited impaired thermogenesis, unexpectedly, they did not have altered *Ucp1* or associated thermogenesis gene expression. Instead, we found that the ALDH2 enzymatic activity is reduced and the 4-HNE level is increased in the BAT from *Aldh2* KI mice. Proteomics screening found that several key mitochondrial proteins involved in mitochondrial fatty acid oxidation (FAO) and electron transfer chain (ETC) were modified by 4-HNE adduction, leading to markedly (~70%) reduced FAO and mitochondrial respiration. Consistently, previous studies have shown that 4-HNE is mainly generated from oxidation of mitochondrial membranes, with 30% of 4-HNE-adducted proteins located within mitochondria[35,36]. We further found the serum free fatty acid level is increased and the respiratory exchange rate (RER) measured by indirect calorimetry is increased in *Aldh2* KI mice, further indicating impaired fatty acid utilization.

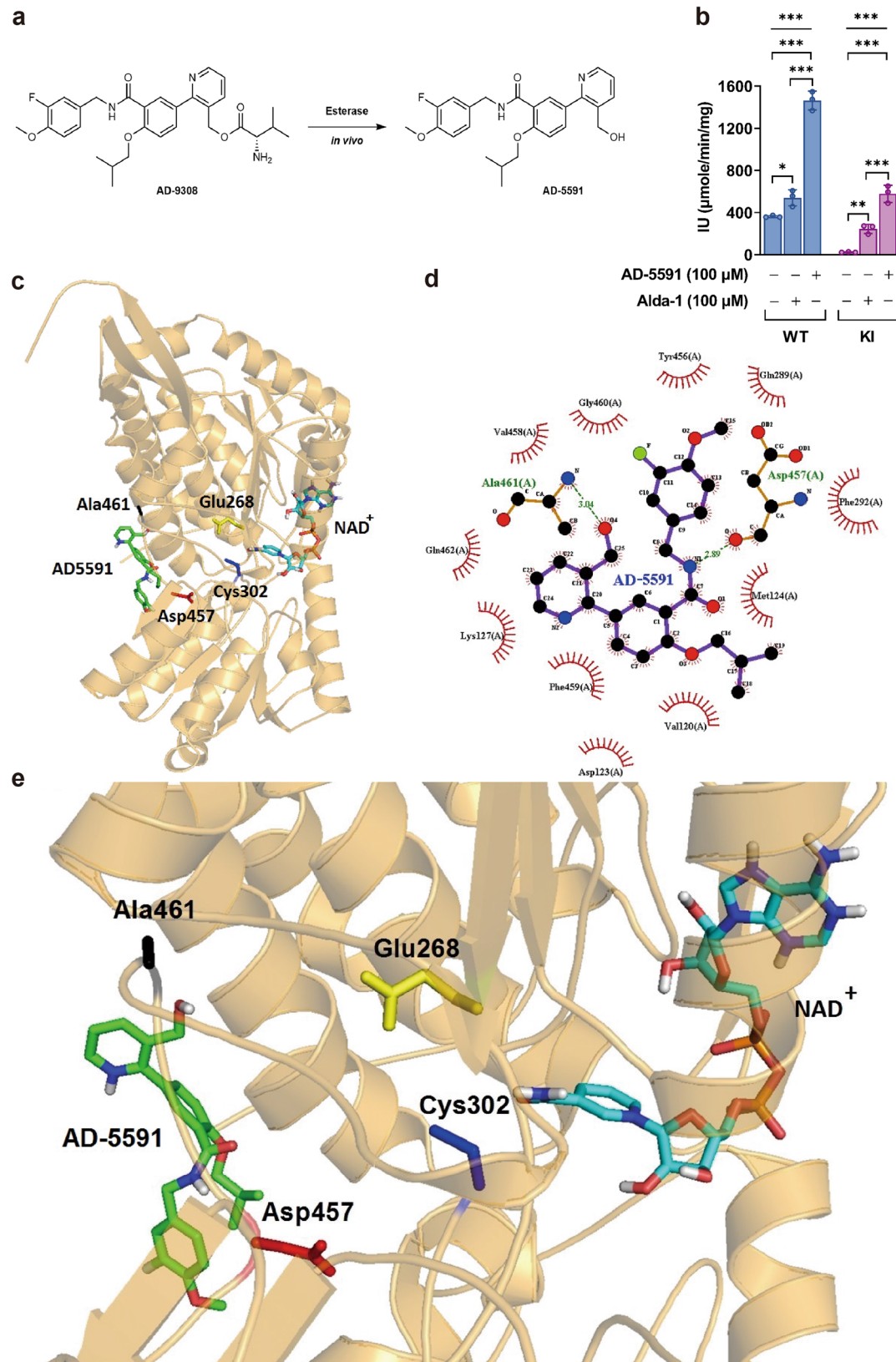

Mitochondrial FAO and ETC are required for the maintenance of the proton gradient in the inter-membranous space, which is essential for Ucp1-mediated adaptive thermogenesis. In our study, we found that the thermogenic capacity of *Aldh2* KI mice was reduced. Fatty acids serve as the main fuel suppliers for thermogenesis[37]. It has been estimated that fatty acids in the BAT contribute 74–84% of the fuel for thermogenesis[37].

Cpt1 is the rate-limiting enzyme for the translocation of fatty acids into mitochondria for β-oxidation. *Cpt1b*[+/−] mice developed fatal hypothermia following cold challenge[38]. Adipose-specific

**Fig. 4 | ALDH2 (Acetaldehyde Dehydrogenase 2) activator AD-9308 enhanced ALDH2 enzymatic activities. a** Prodrug **AD-9308** is converted to **AD-5591** by esterase hydrolysis in vivo. **b AD-5591** (100 µM) or **alda-1** (100 µM) significantly increases the enzymatic activity of recombinant WT and mutant human ALDH2 proteins ($n = 3$ per group in duplicates, left panel: one-way analysis of variance (ANOVA) $P < 0.0001$, Tukey's post hoc test: column 1 vs. 2: $P = 0.04$, column 1 vs. 3: $P < 0.001$, and column 2 vs. 3. $P < 0.001$, right panel: one-way ANOVA $P < 0.0001$, Tukey's post hoc test: column 1 vs. 2: $P = 0.0056$, column 1 vs. 3: $P = 0.0006$, and column 2 vs. 3. $P < 0.001$). **c** Ribbon diagram showing the binding pocket of **AD-5591** within human WT ALDH2. **d** LigPlot (Ligand-Protein Interaction Diagrams) showing the bonds between **AD-5591** and human WT ALDH2. **e** Regional ribbon diagram showing the binding of **AD-5591** and NAD$^+$ with WT ALDH2. Figure (**b**) was analyzed by one-way ANOVA with Tukey's post-hoc tests. All data are presented as mean and standard error (S.E.M.). The $n$ values represent biological repeats and the number of technical repeats were expressed as duplicates or triplicates. The asterisks indicate two-sided *$P < 0.05$, **$P < 0.01$, ***$P < 0.001$.

*Cpt2*-knockout mice presented a hypothermic phenotype when exposed to cold[39]. Mice deficient in fatty acid β-oxidation enzymes, including very-long-chain acyl-CoA dehydrogenase (VLCAD), long-chain acyl CoA dehydrogenase (LCAD), and short-chain acyl CoA dehydrogenase (SCAD) also displayed cold intolerance[40–42]. These data indicate that mitochondrial FAO is critical for adaptive thermogenesis. Furthermore, BAT-specific *Lkb1*-knockout mice, which have reduced expression of ETC complex proteins, also developed impaired thermogenesis[43], indicating that the integrity of mitochondrial ETC machinery is essential for adaptive thermogenesis. These data strongly support our findings that 4-HNE adduction to mitochondrial proteins involved in mitochondrial FAO and ETC could lead to impaired adaptive thermogenesis.

Collectively, these data indicate that the significantly reduced Aldh2 activity of *Aldh2* KI mice results in elevated toxic aldehydes levels and, especially 4-HNE and increased 4-HNE adduction to proteins involved in mitochondrial reduced FAO and mitochondrial respiration of BAT, leading to markedly decreased FAO rate and mitochondrial respiration and subsequent reduced adaptive thermogenesis and energy expenditure. The reduced thermogenesis and energy expenditure result in die-induced obesity and associated metabolic disorders including fatty liver, insulin resistance, and glucose intolerance. On the contrary, over-expression of *Aldh2* has been shown to decrease both the heart and liver weight and mitochondrial injury in mice fed on a high-fat diet[44].

ALDH2 metabolizes bioactive toxic aldehydes by oxidation. In addition to ALDH2, one of the major pathways to detoxify 4-HNE is mediated through glutathione transferases (GSTs) by conjugation to glutathione. Gsta4 is one of the isoforms of GST with the highest conjugation activity for 4-HNE. Consistent with the present finding, disruption of *Gsta4* in mice increased 4-HNE levels and caused obesity in mice[45]. Disruption of the *Gst-10* gene, which causes a 50% increase in 4-HNE adducts in *C.elegans* also resulted in fat accumulation and direct treatment with 4-HNE increases lipid storage in *C.elegans*[46]. Conversely, over-expression of *Gst-10* led to 4-HNE reduction and a lean phenotype[46]. Glutathione peroxidase 4 (*Gpx4*), which resides in the inner mitochondrial membrane, is a scavenger that reduces lipid peroxides. Deficiency of *Gpx4* in mice increased the number of 4-HNE adducts and exacerbated glucose intolerance, dyslipidemia, and fatty liver[47].

Our study also showed that the *Aldh2* KI mice had more severe hepatic steatosis than the WT mice when fed on HFHSD, which can be reversed by **AD-9308** treatment. In line with the present results, **Alda-1**, a prototype of the ALDH2 activator has been shown to alleviate nonalcoholic hepatic steatosis in apolipoprotein E-knockout mice[48] and reversed alcohol-induced hepatic steatosis in animals[49], supporting activation of ALDH2 also prevents both alcohol and non-alcoholic hepatic steatosis.

Our study has several limitations. First, obesity and related metabolic phenotypes in the *Aldh2* KI mice were observed only when fed on HFHSD and not when placed on chow diet, the metabolic phenotypes were not different between the *Aldh2* KI and WT mice. Therefore, our findings may not be generalized to the entire population. The HFHSD for mice consists of 58% calories from fat and 12% calories from sucrose; while the chow diet is composed of 13% calories from fat and 3% calorie from sucrose. According to the Nutrition and

Health Surveys in Taiwan (NAHSIT) in 2008, the average calorie intake from fat and sucrose is 33% and 8% in adults in Taiwan[50]. In the National Health and Nutrition Examination Survey of U.S. adults, the average calorie intake from fat and sucrose is 35% and 14% in 2012[51,52]. The sucrose content is comparable between HFHSD for mice and modern human diet but the fat content of HFHSD is higher than human diet. Even if this concern is relevant, the findings of this study are still substantial given the large number of East Asians (560 million people) carrying this inactivating mutation (Glu504Lys) in the *ALDH2* gene. Second, although the metabolic disorders were normalized in mice treated with **AD-9308** for 5 months without pathological changes in the liver and kidney, long-term safety should be formally determined with GLP standard. Last, humans and other mammals have 19 different aldehyde dehydrogenases (ALDH) and at least six are found in the mitochondria[7]. Although **AD-9308** does not activate ALDH1A1, ALDH3A1, ALDH4A1, ALDH5A1 and ALDH7A1 members of the ALDH family, it is possible that **AD-9308** may still have non-specific effect on other ALDHs.

Our study has important clinical implications. We showed that reduced activity of the mitochondrial enzyme, ALDH2, exacerbates obesity-associated pathologies. These pathologies correlate with increased aldehydic load and inactivation of critical mitochondrial proteins involved in FAO and ETC. Significantly, we showed that treatment with an activator of ALDH2, such as **AD-9308** prevented these pathologies. These data provide strong evidence for a critical role of toxic aldehydes accumulation and defective ALDH2 activity in the pathogenesis of obesity, diabetes, and fatty liver disease. Our study provides a feasible strategy by targeting ALDH2 for the treatment of obesity-associated metabolic disorders which are rising rapidly in human populations, particularly in the East and South Asia.

## Methods
### Animal experiments, administration of diets and drug treatment
All animal experiments were performed according to institutional ethical guidelines and were approved by the Institutional Animal Care and Use Committee (IACUC) (IACUC Approval No: 20110418) of the National Taiwan University College of Medicine and College of Public Health, which is accredited by the Association for Assessment and Accreditation of Laboratory Animal Care International (AAALAC).

*Aldh2* KI mice with C57BL6/J background are kind gifts from Dr. Daria Mochly-Rosen of the Stanford University (Supplementary Fig. 17) and **AD-9308** is a kind gift from Dr. Wenjin Yang from the Foresee Pharmaceuticals, Co., Ltd. The mice were housed at 22–24 °C with light: dark cycles of 12:12 h. The mice were fed either a HFHSD (cat. no. D12331, Research Diets) which provided 58% kilocalorie from fat and 12.5% kilocalorie from sucrose or a chow diet (cat no. 5001, Lab Diet). Mice were bred by mating heterozygous mating. For the animal experiments, **AD-9308** was dissolved in water and was delivered daily to mice by oral gavage. Only male mice were used in this study. Mice were anesthetized using inhaled 1.5–2.0% isoflurane (NDC 66794-017-25, Piramal) at a flow rate of 2 liter/min.

### Glucose and insulin tolerance test
Oral and intraperitoneal GTT were evaluated after 6 h of fasting at the age of 24 weeks. Tail blood glucose was collected at 0, 15, 30, 45, 60,

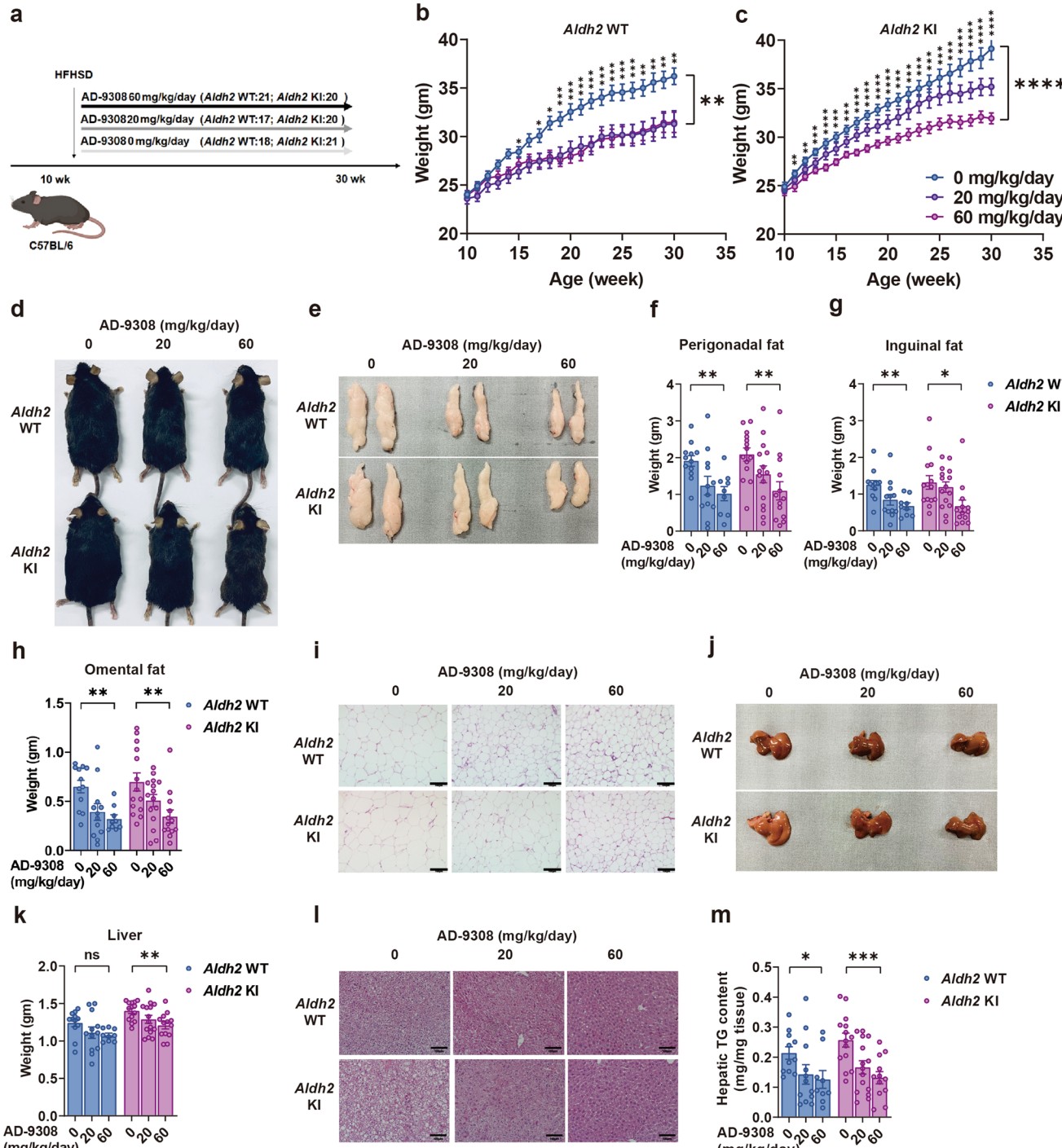

**Fig. 5 | ALDH2 (Acetaldehyde Dehydrogenase 2) activator AD-9308 treatment prevented diet-induced obesity and fatty liver. a** Study flow of **AD-9308** treatment. Body weight of **b** *Aldh2* wild-type (WT) and **c** knock-in (KI) mice treated with vehicle, low-dose **AD-9308** (20 mg/kg/day), and high-dose **AD-9308** (60 mg/kg/day) (repeated measures analysis of variance [ANOVA] $P = 0.0051$ and $P < 0.0001$). **d** Gross appearance of the mice and **e** perigonadal fat of *Aldh2* WT and KI mice treated with **AD-9308**. Weight of **f** perigonadal fat (*P*-for-trend = 0.0048 and 0.0043), **g** inguinal fat (*P*-for-trend = 0.003 and 0.01), and **h** omental fat (*P*-for-trend = 0.0023 and 0.0024) of the *Aldh2* WT and KI mice treated with **AD-9308**. **i** H&E (hematoxylin and eosin) stain of perigonadal fat. The scale bar of is 100 μm. **j** Gross appearance and **k** weight of liver of *Aldh2* WT and KI mice treated with **AD-9308** (*P*-for-trend = 0.06

and 0.0032). **l** H&E stain of liver and **m** hepatic triglycerides (TG) content of the *Aldh2* WT and KI mice treated with **AD-9308** on high-fat high-sucrose diet (*n* = 11:14 for the vehicle group; *n* = 12:15 for 20 mg/kg/day group; *n* = 9:12 for 60 mg/kg/day group in duplicates; *P*-for-trend = 0.039 and 0.0004). The scale bar of is 100 μm. Figures (**b** and **c**) were analyzed using repeated-measures ANOVA. Figures (**f**–**h**, **k**, and **m**) were analyzed using were analyzed using tests for linear trends. All data are presented as mean and standard error (S.E.M.). The *n* values represent biological repeats and the number of technical repeats were expressed as duplicates or triplicates. The asterisks indicate two-sided *$P < 0.05$, **$P < 0.01$, ***$P < 0.001$, ****$P < 0.001$.

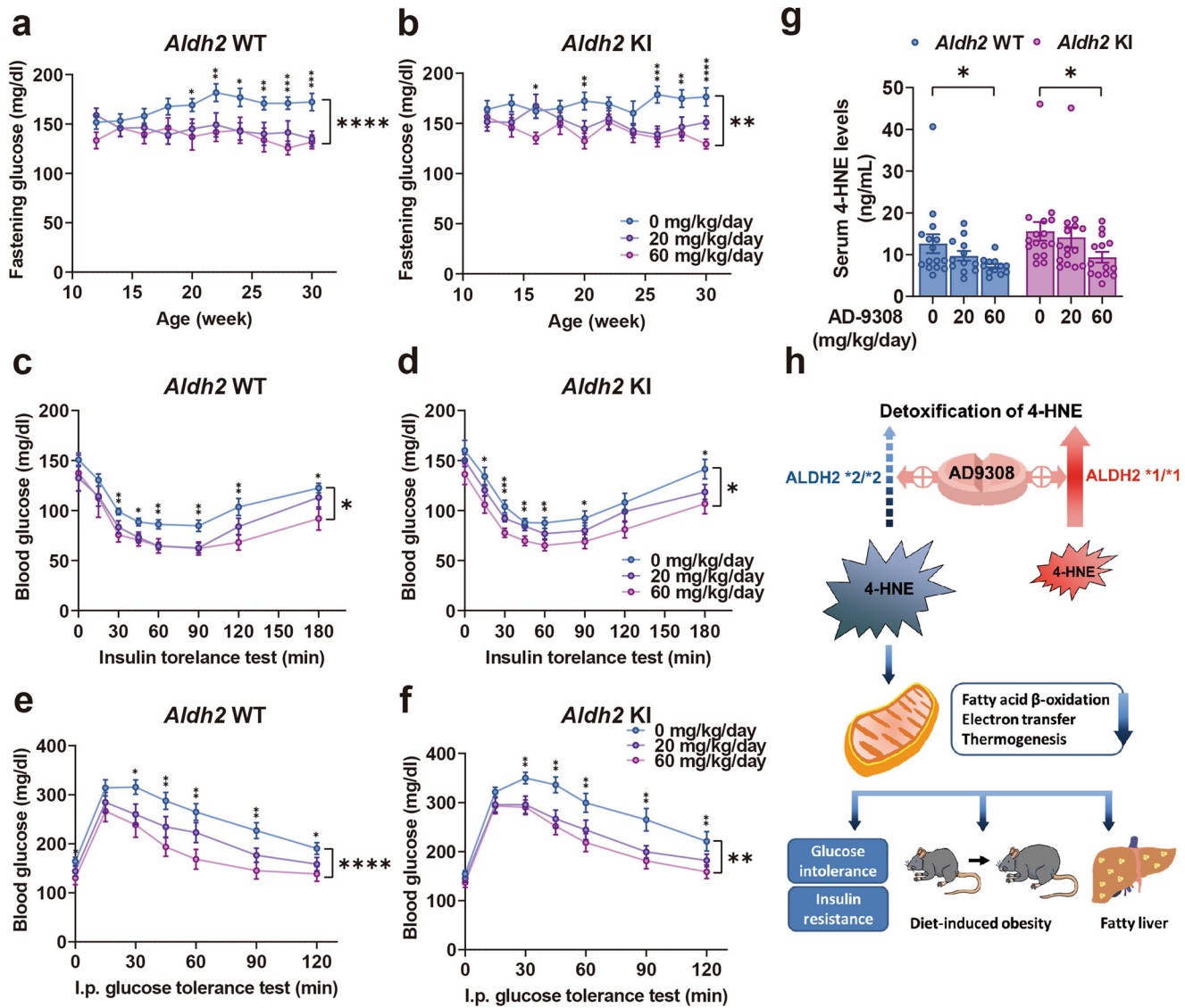

**Fig. 6 | ALDH2 activator AD-9308 treatment ameliorated diet-induced insulin resistance and glucose intolerance.** Fasting glucose level of the **a** *Aldh2* wild-type(WT) and **b** knock-in (KI) mice treated with vehicle, low-dose **AD-9308** (20 mg/kg/day), and high-dose **AD-9308** (60 mg/kg/day) (repeated measures analysis of variance [ANOVA] *P* < 0.0001 and 0.0016). **c**, **d** Glycemic levels during the insulin sensitivity test (repeated measures ANOVA *P* = 0.014 and 0.012) and **e**, **f** intraperitoneal glucose tolerance test of the *Aldh2* WT and KI mice treated with **AD-9308** (*n* = 15:15 for the vehicle group; *n* = 14:16 for 20 mg/kg/day group; *n* = 9:14 for 60 mg/kg/day group; repeated measures ANOVA *P* < 0.0001 and 0.0039). **g** Serum 4-HNE (4-hydroxynonenal) levels of the *Aldh2* WT and KI mice treated with

**AD-9308** (*n* = 15:16 for the vehicle group; *n* = 12:14 for 20 mg/kg/day group; *n* = 12:14 for 60 mg/kg/day group in duplicates; P-for-trend = 0.025 and 0.044) on high-fat high-sucrose diet. **h** Summary diagram depicting how reducing 4-HNE by **AD-9308** ameliorates metabolic disturbances. Figures (**a**–**g**) were analyzed using tests for linear trends. Figures (**a**–**e**) were further analyzed using repeated measures ANOVA. All data are presented as mean and standard error (S.E.M.). The *n* values represent biological repeats and the number of technical repeats were expressed as duplicates or triplicates. The asterisks indicate two-sided \*P < 0.05, \*\*P < 0.01, \*\*\*P < 0.001, \*\*\*\*P < 0.001.

90, and 120 min after oral gavage or intraperitoneal injection of glucose water (1 g/kg) and measured by a glucometer (ACCU-CHECK Performa, Roche). For the ITT, mice were intraperitoneally injected 1 U/kg insulin (Humulin R, Eli Lilly) after 4 h of fasting. Tail blood was collected at 0, 15, 30, 45, 60, 90, 120, and 180 min after injection.

## Energy expenditure, food intake, physical activity, and animal behavior

Metabolic measurements (food and water intake, locomotor activity, $VO_2$ consumption and $VCO_2$ production) were obtained using the Columbus Instruments' Comprehensive Lab Animal Monitoring System. (CLAMS-HC)at the age of 8 weeks. Monitoring was performed for 3 days after mice have been acclimatized to the cages for 3 days. Animal behavior including awakening, drinking, feeding, grooming,

hanging, resting, twisting, walking, and rearing up were recorded by the Clever HomeCage Scan system 3.0 for 24 h after acclimation for 3 days. Accumulative diet intake was measured by weighting diet consumed by two mice of the same genotype in their home cage weekly for 3 weeks.

## Body composition analysis
Body composition was analyzed using the Bruker minispec Live Mice Analyzer (LF50) based on time domain nuclear magnetic resonance technology (TD-NMR) at the age of 24 weeks. The tissue contrast is high between fat and muscle based on relative relaxation times. It acquires and analyzes TD-NMR signals from all protons in the entire sample volume and can provide fat, free and total fluid and lean tissue values for whole-body composition of live, un-anesthetized mice. The

measure frequency is 7.5 Hertz. The accuracy is within 1 % and each measurement was performed in triplicates.

## Isoproterenol-induced energy expenditure

Isoproterenol (20 mg/kg) was dissolved in normal saline with 1:100 dilution (1 mg/100 μL) at the age of 8–10 weeks. After single injection subcutaneously over the neck, energy expenditure was recorded for 5 h.

## Hepatic triglycerides content measurement

Approximately 80 mg of liver tissue from mice at the age of 24 weeks was homogenized in 1800 μl of chloroform/methanol (2:1). Then, 360 μl of $H_2O$ was added. The homogenates were centrifuged and the lower 200-μl layer was added with 100 μl of chloroform with 4% Triton X-100 and then dried in a chemical hood. The dried pellet was re-dissolved with 200 μl of $H_2O$ for determination of triglyceride concentrations with Wako TG LabAssay kit (cat. no. 290-63701, Wako).

## Cold tolerance test and diet-induced thermogenesis test

For the acute cold tolerance test, 20–24-week-old mice with matched body weight from the two groups were placed individually on HFHSD in a 4 °C chamber for 4 h without food supply. For prolonged cold tolerance test, the rectal temperature of the mice was measured after 0, 1, 2, 3, 4, 5, 6, 12, and 18 h a 4 °C chamber with food and water supply. For the diet-induced thermogenesis test, 24-week-old mice were fasted overnight for 18 h. Then, their rectal temperature was measured at 0, 15, 30, 60, 120, 180, and 240 min after HFHSD re-feeding.

## Real-time quantitative PCR (RT-qPCR)

RT-qPCR was performed using SYBR green reagent (cat. no. 11203ES08, YEASEN). The primer sequences for mouse *Ucp1* and *Ppia* is list in Supplementary Table S1. RT-qPCR reactions were performed using an ABI 7900HT FAST (Applied Biosystems). All qPCR reactions were run in duplicate for each sample.

## Primary brown adipocyte culture

BAT from *Aldh2* WT and KI mice aged 4–6 weeks was minced and digested by type I collagenase (cat. no. 17018029, Thermo Fisher Scientific) in HEPES buffer. The stromal vascular fraction (SVF) was obtained by centrifugation and cultured in Dulbecco's modified Eagle medium: nutrient mixture F-12 (DMEM/F-12) (cat. no. 12500062, HyClone) supplemented with 10% FBS (cat. no. 04-001-1A, Biological Industries) and 1× antibiotic/antimycotic solution (cat. no. SV30079.01, HyClone). For cell differentiation, preadipocytes were cultured in differentiation medium containing 10% FBS, 0.5 mM iso-butyl-methylxanthine, 1 μg/ml insulin, 5 μM dexamethasone, 1 nM T3, and 125 μM indomethacin for 2 days. Next, cells were maintained in the medium containing DMEM/F12 with 10% FBS, 1 μg/ml insulin, and 1 nM T3. The medium was changed every 2 days.

## Isolation of mitochondria from brown adipose tissue

For fatty acid assay, BAT was taken from *Aldh2* WT or KI mice fed a HFHSD for 5 weeks. Isolation of mitochondria was performed according to published protocols[53]. Briefly, BAT was homogenized in 10% w/v of ice-cold STE buffer (0.25 M sucrose, 1 mM EDTA, 10 mM Tris-HCl, pH 7.4) by 10 strokes of the loose-fitting pestle of the Dounce homogenizer. The homogenate was spun at $500 \times g$ and 4 °C for 10 min, and the supernatant was kept as the mitochondrial sample for FAO.

## Fatty acid oxidation (FAO) assay

FAO measurements were performed using labeled ${}^3H$-palmitic acid and ${}^3H_2O$ production was assessed as previously described[54]. Briefly, for FAO rate of differentiated primary cells, the capture of ${}^3H_2O$ was measured after a 2-h incubation with 5 mM palmitate/BSA buffer

including 0.5 μCi [${}^3H$]-palmitate (cat. no. PK-NET043001MC, PerkinElmer) in the presence of 1 mM carnitine. For FAO rate rescued assay of differentiated primary cells, the capture of ${}^3H_2O$ was measured after a 2-h incubation with 5 mM palmitate/BSA buffer including a concentration gradient, 0.5, 0.7 and 1.1 μCi [${}^3H$]-palmitate (cat. no. PK-NET043001MC, PerkinElmer) in the presence of 1 mM carnitine. For FAO rate of whole BAT, the isolated mitochondria were placed in a 24-well plate and added incubated with reaction buffer (100 mM sucrose, 10 mM Tris-HCl, 5 mM $KH_2PO_4$, 0.2 mM EDTA, 80 mM KCl, 1 mM $MgCl_2$, 2 mM L-carnitine, 0.1 mM malate, 0.05 mM coenzyme A, 2 mM ATP, 1 mM dithiothreitol, and 7% BSA/500 μM palmitate/0.01 μCi/μl [${}^3H$]-palmitate) at 37 °C for 60 min. For FAO rate rescued assay of whole BAT, the mitochondria were tested in the same condition as above mention, expect the palmitate concentration was used as a concentration gradient, 0.01, 0.01125, 0.0125 and 0.015 μCi/μl. ${}^3H_2O$ was isolated by oil-water separation with chloroform, methanol and KCl/HCl. The average counts per minute (CPM) were measured using a liquid scintillation counter.

## Biotin hydrazide staining for carbonylated protein

For detection of carbonylated protein, samples were chemically reduced by $NaBH_4$ and then incubated with 0.5 mM EZ-link biotin hydrazide (cat. no.21339, Pierce) for 1 h. After coupling, the samples were separated by 10% SDS-PAGE (sodium dodecyl sulfate poly-acrylamide gel electrophoresis gel) and transferred to PVDF (poly-vinylidene fluoride) membrane. The membrane was blocked with 10% skim milk in PBS containing 0.05% Tween-20 (PBST) at 4 °C overnight and then incubated with streptavidin-HRP (horseradish peroxidase) (1:1000, cat no. 890803, BD Biosciences) for 1 h at room temperature. Chemiluminescence signals were developed with HRP substrate (cat. no. WBLUR0500, Millipore).

## Plasmid construction, expression and purification of human ALDH2 in *Escherichia coli*

*E. coli* BL21 (DE3) (cat no. EC0114, Thermo Scientific) was transformed with pTrcHi-WT ALDH2 and pTrcHi-KI ALDH2[29]. The transformants were cultured and then induced to express recombinant proteins using 0.4 mM IPTG for 16–18 h at 25 °C. The cells were harvested by centrifugation and broken in lysis buffer by sonication on ice. The recombinant ALDH2 protein was purified with NI-NTA resin (cat. no. 88222, Thermo Fisher) following the user manual.

## ALDH2 enzymatic activity assays

ALDH2 activity was measured by monitoring the production rate of NADH (nicotinamide adenine dinucleotide)/min at 340 nm and 25 °C. Enzyme activity was assayed in 100 μl of reaction mixtures containing 50 mM $Na_4P_2O_7$ (pH 9.5), 0.01% BSA, 10 mM $NAD^+$, 50 μM acetaldehyde and recombinant ALDH2 with or without 100 μM **AD-5591** or **Alda-1**. One enzymatic activity unit was defined as the production of 1 μmol NADH per min by 1 mg of human ALDH2 protein.

## Molecular docking

To visualize the interaction between the ALDH2 with activators **Alda-1** and **AD-5591**, the X-ray structure of human ALDH2 (PDB ID: 1NZX) and $NAD^+$ were used. Ligand energy was minimized using the PyRx 0.8 program before docking[55]. Three-dimensional models are generated by using the PyMOL v7.4 program[56] and the 2D protein-ligand interaction diagrams were generated by the LigPlot+ 2.2 program[57].

## LC–MS/MS analysis of 4-HNE-modified proteins

The protein samples were prepared with in-solution trypsin digestion. The peptides were desalted with C18 Zip Tip (Millipore, USA) and dry by vacuum centrifugation. The desalted and dried peptides were resuspended in 0.1% formic acid. LC-MS/MS analysis was performed on a NanoACQUITY UPLC system (Waters, USA) coupled to a

high-resolution mass spectrometer (QE HF-X, Thermo Fisher Scientific). The peptides were injected into a trap column (Symmetry C18, 2 cm × 75 μm i.d.) and then separated in a 25 cm × 75 μm i.d. BEH130 C18 column (Waters, USA) on a gradient from 0% to 85% buffer B (buffer A, 0.1% FA $H_2O$; buffer B, 0.1% formic acid in acetonitrile). The mass spectrometer was operated in data-dependent mode with the following acquisition cycle: an MS scan (m/z 350–1600) recorded at resolution R = 60,000 and MS/MS scans recorded at resolution R = 15,000, which were acquired by HCD (higher-energy C-trap dissociation fragmentation) with collision energy of 28.MS/MS spectra were searched with the Mascot engine (v2.6, Matrix Science) against the UniProtKB mouse protein database using the following parameters: a precursor peptide mass tolerance of 20 ppm and an MS/MS fragment tolerance of 0.02 Da with a maximum of two missed cleavage sites. The following modifications were made to the peptides: static carbamidomethylation on cysteine, variable oxidation on methionine, variable deamidation of asparagine or glutamine, and various 4-HNE modifications on cysteine, histidine and lysine. The cut-off threshold for acquiring significant peptide-to-spectrum matches was $P < 0.05$. The SI table of identified proteins is listed in Supplementary Table S2 and the all MASCOT mass spectrum analyses for identified 4-HNE adducted proteins by liquid chromatography tandem mass spectrometry (LC-MS/MS) were listed the Supplementary Data 1. All raw mass spectrometry data to was uploaded to the MassIVE (Mass Spectrometry Interactive Virtual Environment) database (https://massive.ucsd.edu/ProteoSAFe/static/massive.jsp). The accession code is MassIVE00091724.

### Mitochondrial respiratory chain complex activity assay

The mitochondrial complex spectrophotometric assays were carried out using published protocols[58,59]. Briefly, for mitochondrial complex activity assay and LC-MS/MS analysis, BAT was taken from 30-weeks *Aldh2* WT or KI mice fed a HFHSD and homogenized in ice-cold mitochondrial isolation buffer (210 mM Mannitol, 70 mM sucrose, 1 mM EGTA, 0.5% BSA, 5 mM HEPES, pH7.2) by 10 strokes of the loose-fitting pestle of the Dounce homogenizer. The homogenate was spun at 800 × *g* and 4 °C for 10 min, and the supernatant was undergone another centrifugation at 8000 × *g* and 4 °C for 10 min. The pellet was re-suspended in STE buffer and measured protein concentration by Bradford assay. Complex I and complex II activities were measured spectrophotometrically by examining the decrease in absorbance due to the reduction of 2,6-dichlorophenolindophenol (DCPIP) at 600 nm. Activity was expressed in nanomoles of DCPIP reduction/min/mg protein (E = 19.1 mmol$^{-1}$·cm$^{-1}$). Complex III-specific activity and complex IV-specific activity were measured by monitoring the reduction of oxidized cytochrome C (IV) and oxidation of reduced cytochrome C (II) at 550 nm, respectively. The activity is expressed as a nanomole of reduced cytochrome of oxidized cytochrome C /min/mg protein (E = 18.5 mmol$^{-1}$·cm$^{-1}$).

### Immunoblots to detect 4-HNE-adducted proteins and Ucp1, Prdm16, Pgc1α, Dio2, and Cidea

Samples prepared with Laemmli sample buffer and were separated by 10% SDS-PAGE gel and transferred to PVDF membrane. The membrane was blocked with 10% skim milk in PBST and incubated at 4 °C overnight with rabbit anti-4-HNE antibody (1:1000; cat. no. PAB1295, Abnova) and then with secondary antibodies with HRP(1:10000; cat. no. GTX26721, GeneTex). Immunoblots for Ucp1, Prdm16, Pgc1α, Dio2, Cidea, Hsp70, and Gapdh were performed using rabbit polyclonal anti-Ucp1 antibody (1:1000, cat. no. GTX10983, GeneTex), rabbit polyclonal anti-Prdm16 (1:1000, cat. no. ab106410, Abcam), rabbit polyclonal anti-Pgc1α antibody (1:1000, cat. no. NBP1-04676PCP, Novus), rabbit polyclonal anti-Dio2 antibody (1:1000, cat. no. GTX81072, GeneTex), rabbit polycloncal anti-Cidea (1:1000, cat. no. ab8402, Abcam), rabbit monoclonocal anti-Hsp70antibody (1:4000. cat. no.ab45133, Abcam),

and rabbit polyclononal anti-Gapdh antibody (1: 5000,GTX100118, GeneTex).

### Oxygen consumption rate (OCR)

Stromal vascular fraction was isolated from BAT of *Aldh2* WT and KI mice at the age of 4–6 weeks and seeded to Seahorse XF24 v7 cell culture plates (Agilent) at approximate density of 20,000 cells per well and was then differentiated to primary brown adipocyte as described above. Briefly, on day7 of differentiation, cells were washed twice with Seahorse medium and maintained in Seahorse medium in $CO_2$-less incubators. On the day before measurement, the medium was changed to DMEM medium containing 2% lipid-depleted FBS (cat. no. 26400044, Gibco), 3 mM glucose, and 2.5 mM glutamate. Basal OCR was determined by four measurements, followed by three measurements after each injection drug: oligomycin (2 μM), FCCP(carbonyl cyanide-p-trifluoromethoxyphenylhydrazone)(2 μM), Rotenone/Antimycin A (0.5 μM) using Seahorse XFe24 Analyzer (Agilent). OCR measurements of each stage were normalized to the cell number.

### Serum biochemistry

Blood was collected from mice at the age of 24 weeks fasted for 4 h. Fasting serum adiponectin (cat. no. ab108785, Abcam), leptin (cat. no. DY498, R&D), free fatty acid (cat. no. LabAssay 294-63601, Wako), ethanol (cat. no. ab65343, Abcam), norepinephrine (cat. no. ab287789, Abcam), insulin (cat. no. 10-1247-01, Mercodia), and 4-HNE (cat. no. EEL-M2677, Elabscience) concentrations were measured using enzymatic or ELISA assay kits according to manufacturer's instruction. Plasma triglycerides, and total cholesterol levels were measured using the FUJI DRI-CHEM clinical chemistry analyzer.

### Statistical analyses

Two-tailed independent *t*-tests were used for comparing two independent groups. The Wilcoxon rank-sum test was used to compare data from RT-qPCR. Tests for linear trend were used for comparing difference among three genotypes. Data with multiple tests were analyzed with one-way analysis of variance (ANOVA) with Tukey's post-hoc tests. Data with multiple time points were analyzed using repeated measures ANOVA. Energy expenditure among three genotypes was analyzed using the generalized linear model according to international guidance[60,61]. Energy expenditure (expressed as kcal/hr) was regressed on fat and lean mass using the command "glm" implemented in STATA 14.0 and the three genotypes were coded by allelic doses as 0,1, and 2 for WT, HE and KI mice, respectively[60,61]. All data are presented as mean and standard error (S.E.M.). The sample sizes of biological repeats were expressed as "n" and technical repeats were expressed as "in duplicates or triplicates". The asterisks* indicates $P < 0.05$, ** indicates $P < 0.01$, *** indicates $P < 0.001$, and **** indicates $P < 0.0001$. Graphs were generated using GraphPad Prism 9.0. All two-sided *P*-values < 0.05 were considered statistically significant.

### Reporting summary

Further information on research design is available in the Nature Portfolio Reporting Summary linked to this article.

## Data availability

All raw mass spectrometry data was uploaded to the MassIVE (Mass Spectrometry Interactive Virtual Environment) database (https://massive.ucsd.edu/ProteoSAFe/static/massive.jsp). The accession code is MassIVE00091724. The uncropped immunoblot images and source data for all figures are deposited in the Zenodo database https://zenodo.org/record/8262937 and https://doi.org/10.5281/zenodo.8241510. To visualize the interaction between the ALDH2 with activators Alda-1 and AD-5591, the X-ray structure of human ALDH2 (PDB ID: 1NZX) and NAD+ were used (https://www.rcsb.org/

structure/1NZX). All other raw data associated with this study will be provided upon request.

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

## Acknowledgements

We thank the Proteomics Core Facility at Institute of Biomedical Sciences of Academia Sinica and the Metabolomics Core Facility in the Scientific Instrument Center of Academia Sinica for proteomic studies. We thank the support from Taiwan Animal Consortium, Taiwan Mouse Clinic, and the Phenotyping Service and Pathology Core Facility of National Laboratory Animal Center of Taiwan for technical support. We thank Foresee Pharmaceuticals Co., Ltd., Taiwan for the synthesis and provision of **AD-9308** and **AD-5591**. This study was supported by the Ministry of Science and technology of Taiwan (101-2314-B-002-158-MY3, 105-2314-B-002-105-MY3, 106-2321-B-002-040, and 107-2321-B-002 -067) to L.M.C. and Y.C.C., National Taiwan University and National Taiwan University Hospital (UN105-0072 and UN109-008) to L.M.C. and Y.C.C., The New Century Health Care Promotion Foundation Grants to YCC, SPARK Translational Research Program 2013 grant to C.C.H., and NIAAA11147 grant from the NIH to D.M-R. We thank the assistant professor Pi-Hua Liu of the Clinical Informatics and Medical Statistics Research Center of Chang Gung University in Taiwan for statistical assistance. Figure 5a was created with BioRender.com.

## Author contributions

H.L.L. performed the LC-MS/MS analyses, immunoblots, RT-qPCR, histology examinations, and enzyme activity assays. J.Y.H. and W.L.S. performed the animal experiments. Y.C.C and J.Y.N. performed ELISA.Z.Z.D. performed histology examinations. M.L.H. and Y.T.T. conducted the pathological examinations. M.L.H. performed molecular docking. C.C.L. performed the β-oxidation assay and LC-MS/MS experiments. T.Y.L. performed the OCR measurement. W.Y. designed and synthesized AD9308 and AD5591 and conducted pharmacokinetic studies. F.A.L. performed the LC-MS/MS experiments. H.L.C. performed indirect calorimetry. L.Y.C. drew summary graphs. Y.C.C. H.L.L, W.Y., C.H.C. D.M.R. and L.M.C. wrote the manuscript. L.M.C. D.M.R. and Y.C.C. conceived the study and participated in the design and interpretation of all experiments.

## Competing interests

D.M.-R., C.H.C and W.Y. are co-inventors on several issues patents on "Modulators of aldehyde dehydrogenase activity and methods of use thereof", patent Numbers: US 10227304, US 9670162, US 9370506, US 9345693, US 8906942, US 8772295, US 8389522, and US 8354435. W.Y. is an employee and shareholder of Foresee Pharmaceuticals, Co., Ltd. W.Y. is a co-inventor of issued patent US 9879036 "Modulators of aldehyde dehydrogenase activity and methods of use thereof". Other authors declared no competing interests.
