## [Peer Review File · Nature Communications]

A common East-Asian ALDH2 mutation causes metabolic disorders and the therapeutic effect of ALDH2 activatorsREVIEWER COMMENTS

Reviewer #1 (Remarks to the Author):

The paper entitled 'A common East Asian-specific ALDH2 mutation causes obesity and insulin resistance: therapeutic effect of reducing toxic aldehydes by ALDH2 activation' by Yi-Cheng Chang and co-workers explores the behaviour of Aldh2*2/*2 homozygous knock-in (KI) mice, mimicking human Glu504Lys mutation, in developing diet-induced obesity, glucose intolerance, insulin resistance, and fatty liver on a high-fat high-sucrose diet compared with controls.

A key point of the paper is the proteomic analyses of the brown adipose tissue of the Aldh2 KI mice in which an increase of 4-HNE-adducted proteins involved in fatty acid oxidation and electron transport chain has been reported. In the present form, any MS data is consistently presented.

First of all, all the raw data of the LCMSMS analysis has to be loaded on a public proteome identification database to give the opportunity to the reviewers to access to the results using the opportune code. Then, at least in the SI files, all the parameters of the MASCOT analysis (in the original output) have to be reported for all the proteins reported in Figure 3 panel D together with the MSMS spectra of the HNE-modified peptides. A table reporting all the MASCOT results such as Mascot Score, empai, pep matches and sig matches has to be provided, as well.

After this, I will be happy to review this part of the paper.

Another crucial point of the paper is the comparison of the expression levels of many different proteins or protein categories in between the KI and ctr samples. Correctly, in the immunoblotting analysis, the authors show the signal relative to GAPDH and/or HSP70 as loading control. Unfortunately, the loading is not uniform in many lanes of several experiments (such as Figure 3B, S2 and S6) and they don't comment this incongruity. Usually, when it happens, there are two ways to proceed: the researchers can make a densitometric analysis of each signal of immunoblotting and report the intensity of each band taking into account the intensity of the loading control on a histogram or they can reload the samples to obtain a more uniform loading control. The authors have absolutely to ameliorate these data and they have to provide the uncropped western blot analysis in original.

Reviewer #2 (Remarks to the Author):

In this manuscript, Chang and colleagues assessed the role of the East Asian-specific ALDH2 mutation on obesity and glucose metabolism in knock-in mice for this mutation. The authors report that mice that carry the mutation show increased body weight, glucose intolerance and hepatic steatosis. The KI mice further displayed reduced energy expenditure and impaired fatty acid oxidation in BAT. The latter is hypothesized to account for the observed differences in energy expenditure.

The manuscript is well written, easy to understand and deals with an important and innovative research question, the role of the ALDH2 Glu504Lys mutation on energy and glucose metabolism. Using elegant studies, the authors link a well-known human mutation to a functional relevance for energy and glucose metabolism. The manuscript is considered as of importance for the field.

Fig. 1J please show a higher magnification of this image. It's hard to see lipids using the current magnification.

Fig. 2a. here the authors corrected energy expenditure by body weight. This procedure is not commonly accepted anymore since energy expenditure does not increase linear to body weight. The current gold standard is to plot body weight (g, x-axis) against total energy expenditure (kcal/h; Y-axis) followed by ANCOVA analysis using body weight as co-variate (PMID: 23520145; PMID: 23520145). On another note, the authors show in Figure 1A that body weight is significantly higher in the KI mice relative to WT. But in Fig. 2a, (despite sub-optimally corrected), the energy expenditure is only slightly different at 7 (out of 24) time points. I doubt that the total mean energy expenditure will be significantly different. Furthermore, those rather minimal differences at only a few post-hoc time points cannot explain the body weight difference shown in Fig. 1a. Does the ANCOVA give a significant genotype effect? Also, the two peaks in energy expenditure differences occur at the onset of the dark phase (I assume) and the shortly before the light goes on (I assume). Those two occasions typically coincide with increased meal intake in mice (they show typically an increase in food intake at the onset of the dark phase and shortly before the lights go on). So I doubt that energy expenditure differences can really explain the body weight data. Are

genes related to thermogenesis different in BAT of the KI mice (Ucp1, Pgc1a, prdm16, cidea9? Is isoproterenol stimulation of energy expenditure different between wt and KI mice? What is the energy expenditure during acute and chronic cold exposure?

The authors say that the KI mice have no difference in basal metabolic rate. Notably, basal metabolic rate refers to the lowest energy expenditure of a post-absorptive (fasted) animal at thermoneutrality. What the authors describe here is the resting metabolic rate, not basal metabolic rate. RMR is described as the lowest energy expenditure of a resting animal at a given temperature.

Food intake should please be given as cumulative food intake measured over multiple weeks. A simple bar graph is not adequate to rule out cumulative differences over time.

The authors say to have measured cold-induced and diet-induced thermogenesis in Fig. 2f and g. But the authors show here rectal temperature. This is not cold-induced and diet-induced thermogenesis. The authors need to measure energy expenditure in the cold and after feeding a HFD. Assessment of rectal temperature cannot be declared as cold-induced thermogenesis. Also, the observation that rectal temperature is lower in the KI mice does not necessarily mean that also adaptive thermogenesis is different. The mice could also have a conductance phenotype, means they lose more energy over the skin and therefore show lower core temperature. If the mutation e.g. causes vasodilation, then the mice could lose more energy over the surface. So we can't say that lower rectal temperature means that adaptive thermogenesis is different. And the authors report in this figure no data on whether BAT function is impaired. What about expression of key thermogenic genes in BAT (Ucp1, Pgc1a, Prdm16, Cidea)? What about isoproterenol stimulation of energy expenditure? Do BAT primary cells from the KI show differences in adipocyte differentiation (assessment of differentiation marker at different time points of differentiation would be helpfu)? Are catecholamine levels (especially norepinephrine) different in wt vs KI mice?

In Fig. 3a the authors show that Ucp1 expression is not changed in KI mice. What about Ucp1 protein level? What about expression of Pgc1a, Prdm16, cidea? If fatty acid oxidation, and hence fatty acid supply to BAT, is causal for the phenotype, then supplementation of

free fatty acids should reverse this effect. Hence, such a rescue experiment would be valuable to consolidate this hypothesis. Are there also decreased level of FA oxidation in other tissues (liver, muscle etc)? Is the respiratory quotient (RER) different between the genotypes?

What media was used in the seahorse studies of Fig. 3j? Did this media contain FFA? If so, then alterations in fatty acid oxidation can hardly explain the data in Fig. 3J, no?

In Fig. 2 I miss data on FFA. If lower FA oxidation is driving mechanism for the shown phenotype on BAT, the levels of FFA should be different.

In Fig. 5 I miss data on food intake, energy expenditure, RER, locomotor activity, BAT function, FA oxidation, plasma level of FFA, TAG etc. The current figure is lacking quite a few metabolic data that would be helpful to better understand these data

Reviewer #3 (Remarks to the Author):

This manuscript suggested the activation of ALDH2 (by ALDH2 activator) reduces toxic aldehyde levels, and eventually it will be applicable for treating metabolic diseases. Based on the results obtained, the authors claimed that ALDH2 KI (human mutant gene GLu504Lys) were more susceptible to high fat diet high sucrose diet (HFHSD)-induced fatty liver and obesity, and this was due to reduced energy expenditure/impaired thermogenesis and increased aldehyde adduct formation. In addition, ALDH2 KI exhibited reduced insulin sensitivity and impaired glucose tolerance.

<Major points>

- The authors used homozygous ALDH2 KI mice (homozygote $Aldh2^{*2}/^{*2}$) in this study. However, in human, most people with ALDH2 variants are heterozygous $ALDH2^{*2}/^{*1}$. Therefore, heterozygous ALDH2 KI ($Aldh2^{*1}/^{*2}$ knock-in) should be tested to see whether heterozygous $ALDH2^{*2}/^{*1}$ are also more susceptible to HFHSD-induced obesity and metabolic dysfunction.
- The authors clearly demonstrated ALDH2 KI mice are more sensitive to obesity and insulin

intolerance. Based on the results the authors presented, there was no further information what the major factor is to affect mitochondrial fatty acid oxidation, and the production of 4-HNE. One possibility is that long-term high fat diet might change intestinal microbiota changes and has a chance to produce endogenous alcohol (auto-brewery syndrome in human). In addition, high conc. of sugar might accelerate to produce endogenous alcohol, and eventually leads alcohol mediated adduct 4-HNE production. Therefore, the evaluation of blood alcohol concentration (BAC) after HFHSD in WT and ALDH2 KI might be required, and then the authors will have more ideas about the underlying mechanisms.

- In figure1d, the authors need to explain more how to evaluate fat mass, lean mass, free fluids, and total water. Especially, the measurement of free fluids, and total water.

- In some figures, the statistical information was not matched with manuscript. For instance, manuscript Line 104, $P < 0.0001$ (supposed to be ****; Fig1a), however, there are only ** in figure1a.

- Fig1C ($P < 0.01$; it supposed to be **), Fig1i ($P = 0.05$; it supposed to be **). Similar pattern (errors) are been observed in figure set. The authors need to check them very carefully and thoroughly.

- The authors also used “#” in the figure set. What does # mean?

- Line136, the authors evaluated rectal temperature 30 minutes after HFHSD feeding. Could the authors describe the way how to provide HFHSD to experimental mice at the same time for rectal temperature evaluation. Did not the authors provide HFHSD ad-libitum?

- The authors evaluated mouse activity after HFHSD feeding and ALDH2 KI showed grooming behavior was slightly increased. It does not provide any further information/implication, it needs to be removed.

- Provide the information of $Aldh2^{2/2}$ knock-in mouse generation in supporting figure (i.e. sequence data of littermate WT and $Aldh2^{2/2}$ knock-in).

- After treatment of AD-9308, did the authors observe any adverse effects/side effects? Gross/histological changes of organs or serological parameters (such as ALT, AST, BUN, creatinine, bil, amylase and so on).

<Minor points>

- “ $Aldh2$ WT and KI mice” and “ $Aldh2$ KI and WT mice” are mixed in manuscript and figures. It must be written with uniformity

- Many spelling errors (typo) and space error are found. The authors must check them thoroughly throughout the manuscript and figures.

(conten, aldh2*2/*2, Glu504Lysmutation, ALDH2and, Aldh2KI, 60mg/kg/day...)

- Line88, reference number was duplicated 27,27-29

- figure 1e, put BAT, fat, and liver next to images

- Provide full name of MICOS at the first time

- Line200, 201, the expression "good" is not a good choice in original research article. It needs to be revised.

- fig2f, the sample information is missing

- fig4b, the information of X axis is missing

- fig5e,5h, what is the meaning of #?

- In materials and methods, tail blood "glucose" was collected ◇ tail blood was collected

- In materials and methods, please cross-check 58% calorie ◇ 58% kilocalorie, 12.5% calorie ◇ 12.5% kilocalorie

Comments from reviewer

Reviewer #1 (Remarks to the Author):

The paper entitled 'A common East Asian-specific ALDH2 mutation causes obesity and insulin resistance: therapeutic effect of reducing toxic aldehydes by ALDH2 activation' by Yi-Cheng Chang and co-workers explores the behaviour of Aldh2^{*2/*2} homozygous knock-in (KI) mice, mimicking human Glu504Lys mutation, in developing diet-induced obesity, glucose intolerance, insulin resistance, and fatty liver on a high-fat high-sucrose diet compared with controls.

A key point of the paper is the proteomic analyses of the brown adipose tissue of the Aldh2 KI mice in which an increase of 4-HNE-adducted proteins involved in fatty acid oxidation and electron transport chain has been reported. In the present form, any MS data is consistently presented.

(1) First of all, all the raw data of the LC-MS/MS analysis has to be loaded on a public proteome identification database to give the opportunity to the reviewers to access to the results using the opportune code.

Answer: We have uploaded to all raw mass spectrometry data to the ProteomeXchange public dataset base (<https://massive.ucsd.edu/ProteoSAFe/static/massive.jsp>). The accession code is MassIVE00091724.

Then, at least in the SI files, all the parameters of the MASCOT analysis (in the original output) have to be reported for all the proteins reported in Figure 3 panel D together with the MSMS spectra of the HNE-modified peptides.

Answer: We have added all MS/MS spectrum of the identified 4-HNE adducted peptide in **Supplementary Fig. S16** as a separate compressed file.

(2) A table reporting all the MASCOT results such as Mascot Score, empai, pep matches and sig matches has to be provided, as well. After this, I will be happy to review this part of the paper.

Answer: We also have added the SI file in the revised **Supplementary Table 2** and as follows

Table S2. SI table of identified 4-hydroxynonenal (4-HNE)-adducted protein by liquid chromatography tandem mass spectrometry (LC-MS/MS)

Uniprot ID	Protein name	Gene name	Mascot score	Peptide Num.	emPAI	Sequence	Adducted residue (residue site)
Q8BWT1	3-ketoacyl-CoA thiolase, mitochondrial	Acaa2	2275	30	21.75	R.GG K YAVGSACIGGGQGIALIIQNTA.- + HNE (K)	K375
Q9CQB4	Cytochrome b-c1 complex subunit 7	Uqcrb	967	7	13.84	R.DDTL H ETEDVKEAIR.R + HNE (H)	H39
A0A0A6YVZ0	Cytochrome b-c1 complex subunit 1, mitochondrial	Uqcrc1	133	2	9.36	-.DE K NNGAGYFLEHLAFK.L + HNE (K)	K3
Q9CR68	Cytochrome b-c1 complex subunit Rieske, mitochondrial	Uqcrcs1	466	9	2.49	R.AEVLDST K SSKESSEAR.K + HNE (K)	K101
Q8CAQ8	MICOS complex subunit Mic60	Immt	1829	36	5.49	R.KAVDEAADALL K .A + HNE (K)	K296
E9Q800	MICOS complex subunit Mic60	Immt	1826	35	6.54	R.KAVDEAADALL K .A + HNE (K)	K218
Q03265	ATP synthase subunit alpha, mitochondrial	Atp5f1a	1118	22	3.58	R.RVGL K APGIIIPR.I + HNE (K)	K175
P63017	Heat shock cognate 71 kDa protein	Hspa8	281	7	0.5	R.IINEPTAAAIAYGLD K .K + HNE (K)	187
Aldh2 wild-type mice							
Uniprot ID	Protein name	Gene name	Mascot score	Peptide Num.	emPAI	Sequence	Adducted residue (residue site)
D3Z0L4	MICOS complex subunit	Chchd3	555	11	7.88	R.VTFEADENENITV V K.G + HNE (K)	K24
D3Z6F5	ATP synthase subunit alpha	Atp5a1	1067	22	4.28	R.RVGL K APGIIIPR.I + HNE (K)	K125

2. Another crucial point of the paper is the comparison of the expression levels of many different proteins or protein categories in between the KI and ctr samples. Correctly, in the immunoblotting analysis, the authors show the signal relative to GAPDH and/or HSP70 as loading control. Unfortunately, the loading is not uniform in many lanes of several experiments (such as Figure 3B, S2 and S6) and they don't comment this incongruity. Usually, when it happens, there are two ways to proceed: the researchers can make a densitometric analysis of each signal of immunoblotting and report the intensity of each band taking into account the intensity of the loading control on a histogram or they can reload the samples to obtain a more uniform loading control. The authors have absolutely to ameliorate these data and they have to provide the uncropped western blot analysis in original.

Answer: We have added the densitometry analyses and histograms of the immunoblots in the revised manuscript. Please also see revised **Fig. 3b, 3h**, and **Supplementary Fig. S13** as follows. Because it is difficult to present all uncropped immunoblots in the main figures, we prepared a separate compressed supplementary file including all original uncropped immunoblot images and bright field images in the **Supplementary Fig. S16** (compressed file).

“

Figure. 3. (a) Immunoblots and densitometry histogram showing expression of *Ucp*, *Prdm16*, *Cidea*, *Dio2* levels (n=4:4) and (b) densitometric histogram of brown adipose tissue (BAT) from *Aldh2* wild-type (WT) and knock-in (KI) mice.

Fig. 3. (f) Protein carbonylation of BAT mitochondria by hydrazide biotin staining (n=4:4) and (g) immunoblots and densitometry histogram of 4-hydroxynonenal (4-HNE)-adducted mitochondrial proteins in the BAT from the *Aldh2* wild-type (WT) and knock-in (KI) mice on HFHSD (n=3:3). (h) densitometric histogram of (f) and (g).

Supplementary Figure. S13.

Figure. S13. (a) Immunoblots and (b) densitometry histogram showing the *Aldh2* expression in the brown adipose tissue (BAT) from chow-fed and high-fat high-sucrose (HFHSD)-fed C57BL6/J mice.”

Reviewer #2 (Remarks to the Author):

In this manuscript, Chang and colleagues assessed the role of the East Asian-specific ALDH2 mutation on obesity and glucose metabolism in knock-in mice for this mutation. The authors report that mice that carry the mutation show increased body weight, glucose intolerance and hepatic steatosis. The KI mice further displayed reduced energy expenditure and impaired fatty acid oxidation in BAT. The latter is hypothesized to account for the observed differences in energy expenditure.

The manuscript is well written, easy to understand and deals with an important and innovative research question, the role of the ALDH2 Glu504Lys mutation on energy and glucose metabolism. Using elegant studies, the authors link a well-known human mutation to a functional relevance for energy and glucose metabolism. The manuscript is considered as of importance for the field.

1. Fig. 1J Provide a higher magnification image (Fig 1J)

Answer: We have provided an image with higher magnification in the revised **Fig. 1J**.

2. Fig. 2a. here the authors corrected energy expenditure by body weight. This procedure is not commonly accepted anymore since energy expenditure does not increase linear to body weight. The current gold standard is to plot body weight (g, x-axis) against total energy expenditure (kcal/h; Y-axis) followed by ANCOVA analysis using body weight as co-variate (PMID: 23520145; PMID: 23520145).

Answer: We fully agree with the reviewer that ANCOVA or generalized linear model is needed for accurate estimation of metabolic rate because the metabolic rates of different organs, especially white adipose tissue, are different. Nevertheless, an international guide for measuring metabolic rates (Matthias H Tschöp, et al. *Nature Methods* 2012; 9:57-63) recommended that measurement of energy expenditure should be initiated early in the life when body weight is still identical or minimally different so that further statistical correction is not necessary.

As stated in the Methods of our previous manuscript, we used mice with nearly identical body weight at the age of 10-12 weeks for every metabolic rate measurement and core temperature measurement in our study. The measured metabolic rates were further corrected by lean mass. We think the problem caused by different body weights (and different organs with different metabolic rates) will be minimized by this approach.

3. On another note, the authors show in Figure 1A that body weight is significantly higher in the KI mice relative to WT. But in Fig. 2a, (despite sub-optimally corrected), the energy expenditure is only slightly different at 7 (out of 24) time points. I doubt that the total mean energy expenditure will be significantly different. Furthermore, those rather minimal differences at only a few post-hoc time points cannot explain the body weight difference shown in Fig. 1a. Does the ANCOVA give a significant genotype effect? Also, the two peaks in energy expenditure differences occur at the onset of the dark phase (I assume) and the shortly before the light goes on (I assume). Those two occasions typically coincide with increased meal intake in mice (they show typically an increase in food intake at the onset of the dark phase and shortly before the lights go on). So I doubt that energy expenditure differences can really explain the body weight data.

Answer: Based on comments of reviewer 3, we performed additional experiments to see whether *Aldh2* heterozygous knock-in (HE) mice are also more susceptible to diet-induced obesity. We repeated the energy

expenditure measurement of *Aldh2* KI, HE, and WT mice using CLAMS system, as follows (n=19:27:13) using mice with nearly identical body weight. The energy expenditure of *Aldh2* WT mice is significantly higher than KI and HE mice, which can explain the weight difference between three genotypes. Please see revised **Figure 2a** and as follows:

Fig. 2. (a) Energy expenditure of *Aldh2* wild-type (WT), heterozygous knock-in (HE), and homozygous knock-in mice (KI) (n=19:27:13).

Methods

Energy expenditure

Metabolic measurements (food and water intake, locomotor activity, VO_2 consumption and VCO_2 production) were obtained using the Columbus Instruments' Comprehensive Lab Animal Monitoring System (CLAMS-HC). Monitoring was performed for 3 days after mice have been acclimatized to the cages for 3 days.”

4. Are genes related to thermogenesis different in BAT of the KI mice (*Ucp1*, *Pgc1a*, *prdm16*, *cidea9*)?

Answer: We have performed the immunological blots and RT-qPCR for these genes between BAT of WT and KI mice and found no difference in expression of all these thermogenesis genes. Please see revised **Fig.3a&b, Supplementary Fig.S6&S8** and as follows

“

Supplementary Figure 6.

Supplementary Figure. S6. Expression levels of genes involved in thermogenesis including *Ucp1*, *Pgc1α*, *Prdm16* and *Cidea* between in BAT using and real-time quantitative PCR (RT-qPCR)”

Supplementary Figure S8

Please refer to reply to **Point 11**.

5. Is isoproterenol stimulation of energy expenditure different between WT and KI mice?

Answer: We have performed isoproterenol-stimulated energy expenditure for mice of three genotypes (dose: 20 mg/kg, single subcutaneous injection). A significantly increase in energy expenditure was found in wild-type mice (WT) as compared to *Aldh2* homozygous knock-in (KI) and heterozygous knock-in (HE) mice. Please see the revised **Fig. 2h** and as follows:

Figure 2. (h) Isoproterenol-stimulated energy expenditure of *Aldh2* wild-type (WT), heterozygous knock-in (HE), and homozygous knock-in mice (KI) (n=4:8:4). The arrow head indicating the injection time

Methods

Isoproterenol-induced energy expenditure

Isoproterenol (20mg/kg) was dissolved in normal saline with 1:100 dilution (1 mg/100 μ L). After single injection subcutaneously over the neck, energy expenditure was recorded for 5 hr.”

6. What is the energy expenditure during acute and chronic cold exposure?

Answer: Please refer to the answer for **Point 9**.

7. Fig. 2A The authors say that the KI mice have no difference in basal metabolic rate. Notably, basal metabolic rate refers to the lowest energy expenditure of a post-absorptive (fasted) animal at thermoneutrality. What the authors describe here is the resting metabolic rate, not basal metabolic rate. RMR is described as the lowest energy expenditure of a resting animal at a given temperature.

Answer: We thank the review’s correction have corrected the term “ basal metabolic rate” to “resting metabolic rate”.

8. Fig. 2B Food intake should please be given as cumulative food intake measured over multiple weeks. A simple bar graph is not adequate to rule out cumulative differences over time.

Answer: We thank the reviewer’s suggestion and measured dietary intake by weighting diet consumed by two mice of the same genotype in their home cage weekly to avoid stress cause by isolation and frequent interruption for accumulative 21 days. Please see revised **Fig. 2b** and as follows:

Figure. 2. (b) accumulative food intake for three weeks (n=8:24:10)

Methods

Accumulative diet intake was measured by weighting diet consumed by two mice of the same genotype in their home cage weekly for 3 weeks ”

9. The authors say to have measured cold-induced and diet-induced thermogenesis in Fig. 2f and g. But the authors show here rectal temperature. This is not cold-induced and diet-induced thermogenesis. The authors need to measure energy expenditure in the cold and after feeding a HFD. Assessment of rectal temperature cannot be declared as cold-induced thermogenesis. Also, the observation that rectal temperature is lower in the KI mice does not necessarily mean that also adaptive thermogenesis is different. The mice could also have a conductance phenotype, means they lose more energy over the skin and therefore show lower core temperature. If the mutation e.g. causes vasodilation, then the mice could lose more energy over the surface. So we can't say that lower rectal temperature means that adaptive thermogenesis is different.

Answer: We fully agree with the reviewer. Therefore, we measured diet-induced and cold-induced core temperature change and energy expenditure (as well as all experiments regarding core temperature and energy expenditure) using mice of the age of 10-12 weeks with nearly identical body weight to avoid the concern of body surface area and conductance according to the recommendation of an international guide for measuring metabolic rates (Matthias H Tschöp, et al. *Nature Methods* 2012; 9:57-63)

The rectal temperatures are significantly and dose-dependently lower in *Aldh2* homozygous knock-in (KI) and heterozygous knock-in (HE) mice compared with wild-type littermates (WT) in diet-induced thermogenesis test and prolonged cold tolerance tests. Please see revised **Figure 2f&h.**

Figure 2 Diet-induced thermogenesis measured by (f) rectal temperature after high-fat high-sucrose (HFHSD) feeding and (g) cold-induced thermogenesis measured by rectal temperature of mice in 4°C condition (n=9:24:10)

There is also dose-dependently lowered diet-induced increase in energy expenditure of *Aldh2* heterozygous knock-in (HE) and homozygous knock-in (KI) mice. Please see revised **Supplementary Fig. S2** and as follows. Because of animal warfare concern of the indirect calorimetry core facility, we are unable to measure prolonged cold-induced energy expenditure.

Supplementary Figure. S2

Supplementary Figure. S2. Diet-induced increased energy expenditure of *Aldh2* wild-type (WT), heterozygous knock-in (HE), and homozygous knock-in mice (KI). Energy expenditure measured by indirect calorimetry after high-fat high-sucrose feeding for 4 hours after overnight fasting.”

There is no or little difference in rectal temperature in acute cold tolerance tests and acute cold-induced energy expenditure. Please see revised **Supplementary Fig. S3** and as follows.

Supplementary Figure 3. (a) Rectal temperature in 4-hr acute cold tolerance test among (n=9:24:10) and (b) Energy expenditure in acute cold tolerance tests of *Aldh2* homozygous (KI) and heterozygous knock-in (HE) mice compared with wild-type (WT) mice (n=6:6:4)”

Our results that there is significant difference in prolonged cold tolerance tests but no or little difference in acute cold tolerance tests among three genotypes of mice are compatible with the point of view of a review article by Jan Nedergaard, et al. In this review article entitled "Nonshivering thermogenesis and its adequate measurement in metabolic studies" (J Exp Biol. 2011;214 [Pt 2]:242-253), the author stated that "in acute cold tolerance test when mice is acutely transferred to 5°C, they are forced to rely on shivering to defend its body temperature. The capacity and endurance of the shivering is unable to last for long time, and gradually non-shivering thermogenesis will take over". These statements and our finding suggested that acute cold tolerance test is not adequate measurement of non-shivering thermogenesis.

9. And the authors report in this figure no data on whether BAT function is impaired. What about expression of key thermogenic genes in BAT (*Ucp1*, *Pgc1a*, *Prdm16*, *Cidea*)?

Answer: Please refer to the answer for **Point 4**.

10. What about isoproterenol stimulation of energy expenditure?

Answer: Please refer to reply to **Point 5**.

11. Do BAT primary cells from the KI show differences in adipocyte differentiation (assessment of differentiation marker at different time points of differentiation would be helpful)?

Answer: Using primary brown adipocyte isolated from *Aldh2* WT and (KI) mice, we found there is no difference in morphology of brown adipogenesis and expression of *Ucp1*. Please see revised Supplementary Fig. 8 and follows

Supplementary Figure S8.

Figure S8. (a) Morphology showing the differentiation and (b) expression (c) and densitometric histogram of *Ucp1* of primary brown adipocytes isolated from *Aldh2* knock-in (KI) and wild-type (WT) mice”

12. Are catecholamine levels (especially norepinephrine) different in WT vs KI mice?

Answer: We thank the reviewer’s suggestions. The serum norepinephrine levels are not different between three genotypes. Please see revised **Supplementary Fig. S4** and as follows:

“**Supplementary Figure. S4**

Supplementary Figure. S4. Serum norepinephrine levels of *Aldh2* homozygous (KI) and heterozygous knock-in (HE) mice compared with (wild-type) WT mice (n=9:10:11).

Methods

Blood was collected from mice fasted for 4 hours. Fasting serum norepinephrine concentrations (cat. no. ab287789, Abcam) were measured using ELISA kits according to manufacturer's instruction."

13. Fig. 3A What about Ucp1 protein level? What about expression of Pgc1a, Prdm16, cidea?

Answer: Please refer to the answer for **Point 4**.

14. If fatty acid oxidation, and hence fatty acid supply to BAT, is causal for the phenotype, then supplementation of free fatty acids should reverse this effect. Hence, such a rescue experiment would be valuable to consolidate this hypothesis.

Answer: We thank the reviewer's constructive suggestion.

The *ex vivo* FAO rate of the whole BAT tissue isolated from *Aldh2* KI mice was significantly decreased by 70.0% ($P < 0.001$) ($P = 0.02$) as compared to WT mice, which could be rescued by addition of 0.1, 0.2, 0.4, and 0.6 μCi ^3H -palmitate dose-dependently (P -for-trend=0.02) (**Fig. 3k**).

Consistent with the whole BAT tissue, the FAO rate of the cultured primary brown adipocytes isolated from the *Aldh2* KI mice was also significantly decreased by 39.3% ($P = 0.02$) compared with WT mice, which could be rescued by addition of 0.2 and 0.6 μCi ^3H -palmitate dose-dependently (P -for-trend=0.045) (**Fig. 3l**). Please see revised **Fig. 3k&3l** and as follows:

Figure 3. (k) Fatty acid oxidation (FAO) rate of the whole BAT tissue isolated from the *Aldh2*WT and KI mice rescued with 0.1, 0.2, and 0.4 μCi ³H-palmitate respectively (n=5:7:7:7). **(l)** FAO of primary brown adipocytes isolated from the *Aldh2*WT and KI mice rescued with 0.2 and 0.6 μCi ³H-palmitate respectively (n=3:3:3:4)

Methods

Fatty acid oxidation (FAO) assay

FAO measurements were performed using labeled ³H-palmitic acid and ³H₂O production was assessed as previously described⁵⁴. For FAO rate of differentiated primary cells, the capture of ³H₂O was measured after a 2-hour incubation with 5mM palmitate/BSA buffer including 0.5 μCi [³H]-palmitate (cat. no. PK-NET043001MC, PerkinElmer) in the presence of 1 mM carnitine. For FAO rate rescued assay of differentiated primary cells, the capture of ³H₂O was measured after a 2-hour incubation with 5mM palmitate/BSA buffer including a concentration gradient, 0.5, 0.7 and 1.1 μCi [³H]-palmitate (cat. no. PK-NET043001MC, PerkinElmer) in the presence of 1 mM carnitine. For FAO rate of whole BAT, the isolated mitochondria were placed in a 24-well plate and added incubated with reaction buffer (100 mM sucrose, 10 mM Tris-HCl, 5 mM KH₂PO₄, 0.2 mM EDTA, 80 mM KCl, 1 mM MgCl₂, 2 mM L-carnitine, 0.1 mM malate, 0.05 mM coenzyme A, 2 mM ATP, 1 mM dithiothreitol, and 7% BSA/500 μM palmitate/0.01 μCi/μl [³H]-palmitate) at 37°C for 60 min. For FAO rate rescued assay of whole BAT, the mitochondria were tested in the same condition as above mention, except the palmitate concentration was used as a concentration gradient, 0.01, 0.01125, 0.0125, 0.015 and 0.0175 μCi/μl. ³H₂O was isolated by oil-water separation with chloroform, methanol and KCl/HCl. The average counts per minute (CPM) were measured using a liquid scintillation counter.”

15. Are there also decreased level of FA oxidation in other tissues (liver, muscle etc)?

Answer: We thank the reviewer’s suggestion. There was no difference in fatty oxidation rate in skeletal muscle and liver as follows and the **Supplementary Fig.S11**.

Supplementary Fig. S11

Supplementary Fig.S11. Fatty acid oxidation of skeletal muscle (quadriceps) (n=9:8) and liver (n=10:8) from *Aldh2* knock-in (KI) and wild-type (WT) mice.

13. Is the respiratory quotient (RER) different between the genotypes?

Answer: We thank the reviewer's constructive suggestion. The respiratory exchange ratio (RER) of *Aldh2* wild-type (KI) mice are higher than wild-type (WT) mice, indicating impaired fatty acid oxidation. Please see revised **Supplementary Fig. 1** and as follows. Taken together with the finding that *Aldh2* KI mice have elevated free fatty acid levels, these data support our notion that *Aldh2* KI mice have impaired fatty acid oxidation.

“Supplement Fig. 1

Figure S1. Respiratory exchange rate (RER) of *Aldh2* homozygous knock-in (KI) and heterozygous (HE) and wild-type (WT) mice (n=19:27:13)

Methods

Metabolic measurements (food and water intake, locomotor activity, VO_2 consumption and VCO_2 production) were obtained using the Columbus Instruments' Comprehensive Lab Animal Monitoring System (CLAMS-HC).”

16. What media was used in the seahorse studies of Fig. 3J? Did this media contain FFA? If so, then alterations in fatty acid oxidation can hardly explain the data in Fig. 3J, no?

Answer: We used fatty-acid free bovine serum albumin (BSA) and lipid-deleted fetal bovine serum (FBS) in all Seahorse assays as well as fatty acid oxidation assay. We have added detailed description for the revised Methods as follows:

“Oxygen consumption rate (OCR)

Stromal vascular fraction was isolated from BAT of 4-week-old *Aldh2* WT and KI mice and seeded to Seahorse XF24 v7 cell culture plates (Agilent) at approximate density of 20,000 cells per well and was then differentiated to primary brown adipocyte as described above. Briefly, on day7 of differentiation, cells were washed twice with Seahorse medium and maintained in Seahorse medium in CO₂-less incubators. On the day before measurement, the medium was changed to DMEM medium containing 2% lipid-depleted FBS (cat. no. 26400044, Gibco), 3mM glucose, and 2.5mM glutamate. Basal OCR was determined by four measurements, followed by three measurements after each injection drug: oligomycin (2μM), FCCP (2μM), Rotenone/Antimycin A (0.5μM) using Seahorse XFe24 Analyzer (Agilent). OCR measurements of each stage were normalized to the cell number.”

17. In Fig. 2 I miss data on FFA. If lower FA oxidation is driving mechanism for the shown phenotype on BAT, the levels of FFA should be different.

Answer: We thank the reviewer’s precious opinion. Consistent with reduced fatty acid oxidation rate observed in *Aldh2* knock-in mice, there is a dose-dependent increase in serum free fatty acid levels among *Aldh2* homozygous knock-in (KI), heterozygous knock-in (HE) mice, and wild-type (WT) mice. Please see revised **Fig 2n**. This data indicates *Aldh2* homozygous knock-in might have impaired fatty acid oxidation rate and /or increased lipolysis. Together with the lower respiratory exchange rate (RER) of *Aldh2* KI and HE compared with WT mice, which indicated lower fatty acid utilization. These data suggest lowered fatty acid utilization.

Figure 2 (n) free fatty acid, and (n=10:10:10), concentration of *Aldh2* homozygous knock-in (KI) and heterozygous knock-in (HE), and wild-type (WT) mice on high-fat high-sucrose diet. *P < 0.05, **P < 0.01, ***P < 0.001

Serum biochemistry

Blood was collected from mice fasted for 4 hours. Fasting serum free fatty acid concentration (cat. no. LabAssay 294-63601, Wako) were measured using enzymatic assay kits according to manufacturer's instruction"

18. In Fig. 5 I miss data on food intake, energy expenditure, RER, locomotor activity, BAT function, FA oxidation, plasma level of FFA, TAG etc. The current figure is lacking quite a few metabolic data that would be helpful to better understand these data

Answer: We fully agree with the reviewer's comments. However, because of the limitation of core facility during the pandemic, lack of enough mice for experiment (the phenotype of *Aldh2* heterozygous knock-in (HE) mice are required for revision), and the lack of enough AD-9308 compound required for AD-9308 treatment (20 weeks), it is very difficult for us to perform similar assays for AD-9308-treated mice.

Reviewer #3 (Remarks to the Author):

This manuscript suggested the activation of ALDH2 (by ALDH2 activator) reduces toxic aldehyde levels, and eventually it will be applicable for treating metabolic diseases. Based on the results obtained, the authors claimed that ALDH2 KI (human mutant gene GLu504Lys) were more susceptible to high fat diet high sucrose diet (HFHSD)-induced fatty liver and obesity, and this was due to reduced energy expenditure/impaired thermogenesis and increased aldehyde adduct formation. In addition, ALDH2 KI exhibited reduced insulin sensitivity and impaired glucose tolerance.

<Major points>

1. The authors used homozygous ALDH2 KI mice (homozygote Aldh2*2/*2) in this study. However, in human, most people with ALDH2 variants are heterozygous ALDH2*2/*1. Therefore, heterozygous ALDH2 KI (Aldh2*1/*2 knock-in) should be tested to see whether heterozygous ALDH2*2/*1 are also more susceptible to HFHSD-induced obesity and metabolic dysfunction.

Answer: We have tested whether heterozygous Aldh2 knock-in mice (HE) are susceptible to HFHSD-induced obesity and metabolic dysfunction in the Figure 1&2 and as follows.

Fig. 1. Body weight of *Aldh2* homozygous knock-in (KI) and heterozygous knock-in (HE), and wild-type (WT) mice on (a) high-fat high sucrose diet (HFHSD)

(n=24:36: 21) and **(b)** Weights of inguinal, perigonadal, and mesenteric fat, the liver, and brown adipose tissue (BAT) of the *Aldh2* WT, HE and KI mice (n=63:24:48). (c) Body composition of *Aldh2* WT, HE and KI mice (n=16:17:22) (d) Gross appearance of mice, BAT, perigonadal fat and liver. (e) Average adipocyte size and (f) adipocyte number in the perigonadal fat pad of the *Aldh2* KI, HE, and WT mice (n=24:14:17). (g) H&E stain of the perigonadal fat in *Aldh2* KI, HE and WT mice. (h) Hepatic triglyceride content of the *Aldh2* KI, HE, and WT mice (n=7:9:8) (i) Representative H&E staining of livers from the *Aldh2* KI, HE, and WT mice. (j) Muscle triglyceride content in the *Aldh2* KI, HE, and WT mice on HFHSD (n=7:8:8). *P < 0.05, **P < 0.01.

Fig. 2. (a) Energy expenditure (n=19:27:13), (b) accumulative food intake for three weeks (n=8:24:10), (c) wheel rotations (n=9:13:5), (d) distance traveled (n=5:20:7), and (e) behaviors monitored by HomeCage

systems (n=5:20:7) of the *Aldh2* homozygous knock-in (KI), heterozygous knock-in KI (HE), and wild-type (WT) mice (f) Diet-induced thermogenesis measured by rectal temperature after high-fat high-sucrose feeding and (g) cold-induced thermogenesis measured by rectal temperature of mice in 4°C condition (n=9:24:10) (h) isoproterenol-induced energy expenditure (n=4:8:4) of *Aldh2* KI, HE, and WT mice. The arrow head indicating the injection time. (i) Glycemic levels during the insulin sensitivity test and (j) intraperitoneal glucose tolerance tests, and (k) glycemic levels during the oral glucose tolerance test of *Aldh2* KI, HE, and WT mice (n=49:24:42). (l) Insulin levels following oral glucose load (n=35:14:44). Fasting serum (m) leptin (n=9:10:11), (n) free fatty acid, and (n=10:10:10), (o) adiponectin (n=10:10:10), and (p) ethanol (n=10:10:10) concentration of *Aldh2* KI, HE and WT mice on high-fat high-sucrose diet. *P < 0.05, **P < 0.01, ***P < 0.001”

2. The authors clearly demonstrated ALDH2 KI mice are more sensitive to obesity and insulin intolerance. Based on the results the authors presented, there was no further information what the major factor is to affect mitochondrial fatty acid oxidation, and the production of 4-HNE (producing more endogenous alcohol due to high conc. of sugar in diet)

Answer: In *ALDH2* knock-in mice mimicking the East Asian-specific E504K mutation, we found markedly decreased enzymatic activity of ALDH2, an enzyme responsible for detoxification of 4-HNE, resulting in increased 4-HNE levels and 4-HNE-adducted proteins in the brown adipose tissue (BAT), which is essential for adaptive thermogenesis and energy expenditure. Using LC-MS/MS, we found increased 4-HNE-adducted proteins, especially those involved in mitochondrial fatty acid oxidation and electron transfer chain in the BAT from the *Aldh2* knock-in mice, leading to markedly decreased (~70%) fatty acid oxidation rate of f BAT, which could be dose-dependently rescued by addition of fatty acid.

Furthermore, the *Aldh2* knock-in mice displayed higher respiratory exchange rate (RER) measured by indirect calorimetry and elevated circulating fatty acids, which indicates lower fatty acid utilization compared with wild-type mice.

Mitochondrial fatty acid oxidation and mitochondrial respiration of BAT are required for the maintenance of the proton gradient in the inter-membranous space, which is essential for Ucp1-mediated adaptive thermogenesis and energy expenditure. There impaired mitochondrial fatty acid oxidation and mitochondrial respiration of *Aldh2* knock-in mice caused decreased adaptive thermogenesis and energy expenditure observed in *Aldh2* knock-in mice, leading to the development of obesity

We have added additional explanation in abstract of the revised manuscript and as follows:

“Abstract

We found elevated 4-HNE levels and increased 4-HNE adducted proteins due to reduced activity of ALDH2 of the brown adipose tissue (BAT) from the *Aldh2* homozygous KI mice. Proteomic analyses of the BAT from the *Aldh2* KI mice identified increased 4-HNE-adducted proteins involved in mitochondrial fatty acid oxidation (FAO) and electron transport chain (ETC), leading to markedly decreased FAO and mitochondrial respiration of BAT, which is essential for adaptive thermogenesis and energy expenditure.

Discussion

BAT is a highly specialized organ enriched in *Ucp1* for adaptive thermogenesis. Although *Aldh2*KI mice exhibited impaired thermogenesis, unexpectedly, they did not have altered *Ucp1* or associated thermogenesis gene expression. Instead, we found that the ALDH2 enzymatic activity is reduced and the 4-HNE level is increased in the BAT from *Aldh2*KI mice. Proteomics screening found that several key mitochondrial proteins involved in mitochondrial fatty acid oxidation (FAO) and electron transfer chain were modified by 4-HNE adduction, leading to markedly (~70%) reduced FAO and mitochondrial respiration. Consistently, previous studies have shown that 4-HNE is mainly generated from oxidation of mitochondrial membranes, with 30% of 4-HNE-adducted proteins located within mitochondria^{35,36}. We further found the serum free fatty acid level is increased and the respiratory exchange rate (RER) measured by indirect calorimetry is increased in *Aldh2*KI mice, further indicating impaired fatty acid utilization.

Mitochondrial FAO and ETC are required for the maintenance of the proton gradient in the inter-membranous space, which is essential for *Ucp1*-mediated adaptive thermogenesis. In our study, we found that the thermogenic capacity of *Aldh2*KI mice was reduced. Fatty acids serve as the main fuel suppliers for thermogenesis³⁷. It has been estimated that fatty acids in the BAT contribute 74-84% of the fuel for thermogenesis³⁷.

Cpt1 is the rate-limiting enzyme for the translocation of fatty acids into mitochondria for β -oxidation. *Cpt1b*^{+/-} mice developed fatal hypothermia following cold challenge³⁸. Adipose-specific *Cpt2*-knockout mice presented a hypothermic phenotype when exposed to cold³⁹. Mice deficient in fatty acid β -oxidation enzymes, including very-long-chain acyl-CoA dehydrogenase (VLCAD), long-chain acyl CoA dehydrogenase (LCAD), and short-chain acyl CoA dehydrogenase (SCAD) also displayed cold intolerance⁴⁰⁻⁴². These data indicate that mitochondrial FAO is critical for adaptive thermogenesis. Furthermore, BAT-specific *Lkb1*-knockout mice, which have reduced expression of ETC complex proteins, also developed impaired thermogenesis⁴³, indicating that the integrity of mitochondrial ETC machinery is essential for adaptive thermogenesis. These data strongly support our findings that 4-HNE adduction to mitochondrial proteins involved in mitochondrial FAO and ETC could lead to impaired adaptive thermogenesis.

Collectively, these data indicate that the significantly reduced ALDH2 activity of *Aldh2*KI results in elevated toxic aldehydes levels and, especially 4-HNE and increased 4-HNE adduction to proteins involved in mitochondrial reduced FAO and mitochondrial respiration of BAT, leading to markedly decreased FAO rate and mitochondrial respiration and subsequent reduced adaptive thermogenesis and energy expenditure. The reduced thermogenesis and energy expenditure result in diet-induced obesity and associated metabolic disorders including fatty liver, insulin resistance, and glucose intolerance.”

References

35. Zhao, Y., et al. Redox proteomic identification of HNE-bound mitochondrial proteins in cardiac tissues reveals a systemic effect on energy metabolism after doxorubicin treatment. *Free. Radic. Biol. Med.* **72**, 55-65 (2014).
36. Poli, G., Schaur R. J., Siems W. G. & Leonarduzzi, G. 4-hydroxynonenal: a membrane lipid oxidation product of medicinal interest. *Med. Res. Rev.* **28**, 569-631 (2008).
37. Labbé, S. M., et al. In vivo measurement of energy substrate contribution to cold-induced brown adipose tissue thermogenesis. *FASEB. J.* **29**, 2046-2058 (2015).”
38. Ji. S., et al. Homozygous carnitine palmitoyltransferase 1b (muscle isoform) deficiency is lethal in the

mouse. *Mol. Genet. Metab.* **93**, 314-322 (2008).

39. Lee, J. Ellis, J. M. & Wolfgang, M.J. Adipose Fatty Acid Oxidation Is Required for Thermogenesis and Potentiates Oxidative Stress-Induced Inflammation. *Cell. Rep.* **10**, 266-279 (2015).

40. Schuler, A.M., et al. Synergistic heterozygosity in mice with inherited enzyme deficiencies of mitochondrial fatty acid beta-oxidation. *Mol. Genet. Metab.* **85**, 7-11 (2005).

41. Thorpe, C. & Kim, J.J. P. Structure and mechanism of action of the Acyl-CoA dehydrogenases 1. *FASEB. J.* **9**, 718-725 (1995).

42. Gregersen, N., et al. Mutation analysis in mitochondrial fatty acid oxidation defects: Exemplified by acyl-CoA dehydrogenase deficiencies, with special focus on genotype-phenotype relationship. *Hum. Mutat.* **18**, 169-189 (2001).

43. Masand, R., et al. Proteome Imbalance of Mitochondrial Electron Transport Chain in Brown Adipocytes Leads to Metabolic Benefits. *Cell. Metab.* **27**, 616-629.e614 (2018).

3. The evaluation of blood alcohol concentration (BAC) after HFHSD in WT and ALDH2 KI might be required

Answer: We thank the reviewer's previous opinion. There is a dose-dependent increase in serum ethanol levels among three genotypes. Please see revised Figure 2p and as follows:

Figure. 2 (p) ethanol (n=10:10:10) concentration of *Aldh2* homozygous knock-in (KI) and heterozygous knock-in (HE), and wild-type (WT) mice

Methods

Blood was collected from mice fasted for 4 hours. Fasting serum ethanol (cat. no. ab65343, Abcam), norepinephrine (cat. no. ab287789, Abcam) concentrations were measured using ELISA kits according to manufacturer's instruction."

4. Fig. 1D Explain more how to evaluate fat mass, lean mass, free fluids, and total water, especially, the measurement of free fluids, and total water.

Answer: We have added methods for measuring body composition in the Methods of the revised manuscript and as follows.

“Body composition analysis

Body composition was analyzed using the Bruker minispec Live Mice Analyzer (LF50) based on time domain nuclear magnetic resonance technology (TD-NMR). The tissue contrast is high between fat and muscle based on relative relaxation times. It acquires and analyzes TD-NMR signals from all protons in the entire sample volume and can provide fat, free and total fluid and lean tissue values for whole-body composition of live, unanesthetized mice. The measure frequency is 7.5 Hertz. The accuracy is within 1 % and each measurement was performed in triplicates.”

5. In some figures, the statistical information was not matched with manuscript.

For instance, manuscript Line 104, P<0.0001 (supposed to be **; Fig. 1A), however, there are only ** in Fig. 1A.Fig1C (P<0.01; it supposed to be **), Fig. 1I (P=0.05; it supposed to be **). Similar pattern (errors) are been observed in figure set. The authors need to check them very carefully and thoroughly. The authors also used “#” in the figure set. What does # mean?**

Answer: We have checked the figures and unified all symbols, i.e. * indicates P<0.05, ** indicates P<0.01; *** indicates P<0.001.

6. Line136, the authors evaluated rectal temperature 30 minutes after HFHSD feeding. Could the authors describe the way how to provide HFHSD to experimental mice at the same time for rectal temperature evaluation? Did not the authors provide HFHSD ad-libitum?

Answer: We fast the mice overnight. After overnight fasting, the mice would eat food simultaneously.

7. The authors evaluated mouse activity after HFHSD feeding and ALDH2 KI showed grooming behavior was slightly increased. It does not provide any further information/implication, it needs to be removed.

Answer: Because the HomeCage behavior system was changed during revision, we repeated the assays and add the data of *Aldh2* heterozygous knock-in (HE) mice, there was no difference in behavior measured by HomeCage system. Please see revised **Fig. 2e** and as follows:

Figure. 2e behaviors monitored by HomeCage systems (n=5:20:7) of the *Aldh2* homozygous knock-in (KI), heterozygous knock-in (HE), and wild-type (WT) mice.

Methods:

Animal behavior including awakening, drinking, feeding, grooming, hanging, resting, twisting, walking, and rearing up were recorded by the Clever HomeCage Scan system 3.0 for 24 hours after acclimation for 3 days.”

8. Provide the information of *Aldh22/*2 knock-in mouse generation in supporting figure (i.e. sequence data of littermate WT and *Aldh2**2/*2 knock-in.**

Answer: We thank the reviewer’s thoughtful consideration and have added the information about the generation of *Aldh2* homozygous knock-in mouse and the sequencing data in the **Supplementary Fig.S15** and as follows

“Generation of *Aldh2* knock-in mice

Aldh2 KI mice carrying the human ALDH2 Glu504Lys mutation were generated by introducing the Glu504Lys mutation within the mouse gene². The details of generation were described previously²⁹. Briefly, *Aldh2* knock-in mice were generated by homologous recombination with a 8.0-kb genomic fragment encompassing the mouse ALDH2 locus carrying a site-directed mutagenesis within exon 12 of the *Aldh2* genomic fragment corresponding to the position of human E487K mutation. The schematic graphic was shown in **Supplementary Fig.S15**. The sequence of specific primers EG475, EG460, and EG399 were used for the amplification of a 1.3-kb fragment from exon 13 to the 3’ untranslated region (UTR) of neomycin marker and a 3.0-kb fragment from exon 13 to downstream of the neomycin marker, respectively, for the the mutated allele. In contrast, for the wild-type allele, a 1.4-kb fragment devoid of the neomycin marker was amplified using EG475 and EG399 primers². The founder mice were backcrossed to the C57BL/6 background for at least nine generations to achieve a homogeneous genetic background. Both *Aldh2* WT controls and *Aldh2* KI mice were littermates from mated heterozygous (HE) mice.

Supplementary Figure S15.

Supplementary Figure. S12. (a) Schematic graphic of the generation of *Aldh2* mutant allele mimicking human Glu504Lys mutation (b) Sanger sequencing of *Aldh2* wild-type and heterozygous knock-in (HE) mice

Supplementary Reference 2.. Zambelli, V. O., Gross, E. R., Chen, C.H., Gutierrez, V. P., Cury Y. & Daria, M-R. Aldehyde dehydrogenase-2 regulates nociception in rodent models of acute inflammatory pain. *Sci. Transl. Med.*6:251ra118 (2014).

9. After treatment of AD-9308, did the authors observe any adverse effects/side effects?

Gross/histological changes of organs or serological parameters (such as ALT, AST, BUN, creatinine, bil, amylase and so on).

Answer: We thank the reviewer's comments.

(1) The gross appearance of mice was shown in revised **Figure 5d** and as follows. The gross appearance of perigonadal fat and liver was shown in **Figure 5e&5j** and the histology of perigonadal fat and liver is shown in **Figure 5i&l** and as follows

Fig. 5. (d) Gross appearance of the mice and (e) perigonadal fat of *Aldh2*WT and KI mice treated with AD-9308. (i) H&E stain of perigonadal fat (j) Gross appearance and (l) H&E stain of liver of *Aldh2*WT and KI mice treated with AD-9308.

Pathological examination of the liver and kidney by contract pathological examination from 4 *Aldh2* homozygous wild-type (WT) and 4 knock-kin (KI) treated with 0, 20 or 60 mg/kg/day of AD-9308 for 20 reported no abnormality as follows and in the revised Supplementary Table 4.

.Supplementary Table S4. Pathological examination of liver and kidney from *Aldh2*KI and WT mice treated with 0, 20 or 60 mg/kg/day of AD-9308 for 20 weeks scored by H&E stain.

Organ	Histopathological finding	Pathological number																							
		WT												KI											
		0 mg/kg				20 mg/kg				60 mg/kg				0 mg/kg				20 mg/kg				60 mg/kg			
		#1	#2	#3	#4	#1	#2	#3	#4	#1	#2	#3	#4	#1	#2	#3	#4	#1	#2	#3	#4	#1	#2	#3	#4
Liver	Fatty change, hepatocyte	3	1	3	3	2	1	1	2	3	2	2	-	1	1	1	2	1	1	2	1	1	1	2	1
	Accumulation, glycogen, hepatocyte	4	2	2	2	2	1	1	4	3	3	2	2	2	1	1	3	2	2	4	3	3	2	1	2
	Extramedullary hematopoiesis	1	1	1	1	1	1	1	1	1	1	1	1	1	1	1	1	1	1	1	1	1	1	1	1
Kidney	Vacuolation, cytoplasmic, renal tubule	2	2	2	2	2	2	2	2	2	2	N	2	2	2	2	2	2	1	2	1	2	1	2	2
	Infiltration, mononuclear cell, interstitium	1	1	1	1	1	1	1	1	1	1	N	1	2	1	1	1	1	1	1	2	2	1	1	1

N: No available tissue. -: No significant lesions. Degree of lesions stained with HE was graded from one to five depending on severity: 1 = minimal (< 1%); 2: slight (1-25%); 3 = moderate (26-50%); 4 = moderate/severe (51-75%); 5 = severe/high (76-100%).

*Contract pathological examination by the Pathology core of the National Animal Research Laboratory, Taiwan

(3) Toxicology parameters including serum alanine aminotransferase (ALT) and creatinine levels of *Aldh2*KI and WT mice treated with 0, 20 or 60 mg/kg/day of AD-9308 for 20 weeks showed no hepatic or renal toxicity. Please see revised Supplementary Figure 14

Supplementary Figure 14.

Supplementary Figure.S14. Serum (a) creatinine and (b) alanine aminotransferase (ALT) and levels of *Aldh2*KI and WT mice treated with 0, 20 or 60 mg/kg/day of AD-9308 for 20 weeks (n=6-7 per group).

Supplementary Methods

Pathological examinations

Pathological examination was performed by the contract pathological core service of the Animal Centers of Medical College, National Taiwan University. Pathological changes of liver and kidney from *Aldh2* KI and WT mice treated with 0, 20 or 60 mg/kg/day of AD-9308 for 20 weeks was examined and scored by H&E stain. Serum alanine aminotransferase (ALT) and creatinine levels of *Aldh2*KI and WT mice treated with 0, 20 or 60 mg/kg/day of AD-9308 for 20 weeks were assayed using the FUJI DRI-CHEM clinical chemistry analyzer.”

<Minor points>

1. “*Aldh2* WT and KI mice” and “*Aldh2* KI and WT mice” are mixed in manuscript and figures. It must be written with uniformity
2. Many spelling errors (typo) and space error are found. The authors must check them thoroughly throughout the manuscript and figures.(content, *aldh2**2/*2, Glu504Lysmutation, ALDH2and, *Aldh2*KI, 60mg/kg/day...)

Answer: We have unified all expression to “*Aldh2* homozygous knock-in (KI), heterozygous knock-in (HE) and wild-type mice (WT)” throughout the manuscript.

3. Line88, reference number was duplicated 27,27-29

Answer: We thank the review for the careful reading and have corrected this error.

4. Fig. 1E, put BAT, fat, and liver next to images

Answer: We have added the label of tissue to the images. Please see revised **Figure 1d** and as follows

Fig 2. (d) Gross appearance of mice, BAT, perigonadal fat and liver.”

5. Provide full name of MICOS at the first time

Answer: We thank for the review’s correction and have added the full name “mitochondrial contact site and cristae organizing system” in the revised manuscript.

6. Line200, 201, the expression “good” is not a good choice in original research article. It needs to be revised.

Answer: We thank the reviewer’s correction and have change the sentence from

“*In vivo*, AD-9308 administration showed a good pharmacokinetic profile in mice when administered orally or intravenously with good bioavailability in rodents and dogs (**Supplementary Table S3**).”

to

” *In vivo*, AD-9308 administration showed a favorable pharmacokinetic profile in mice when administered orally or intravenously with high bioavailability in rodents and dogs (**Supplementary Table S3**).”

7. Fig. 2F, the sample information is missing

Answer: We thank for the reviewer’s correction and have added additional information in the legend of revised **Fig. 2f&g** as follows.

“**Figure. 2 (f)** Diet-induced thermogenesis measured by rectal temperature after high-fat high-sucrose feeding and **(g)** cold-induced thermogenesis measured by rectal temperature of mice in 4°C condition (n=9:24:10)of *Aldh2* homozygous knock-in (KI) and heterozygous knock-in (HE), and WT mice.”

8. Fig. 4B, the information of X axis is missing

Answer:We thank the reviewer’s correction. We have revised the legend of Figure. 2b as follows:

Figure 2 (b) AD-5591(100µM) or alda-1(100µM) significantly increases the enzymatic activity of recombinant WT and mutant human ALDH2 proteins (n=9 for each group).”

9. Fig. 5E, 5H, what is the meaning of #?

Answer: We used # to indicate P<0.1 but we have deleted this symbol in the revised manuscript.

10. In materials and methods, tail blood “glucose” was collected → tail blood was collected.

Answer: We thank the reviewer’s correction and have deleted “glucose” from the sentence.

11. In materials and methods, please cross-check 58% calorie→58% kilocalorie, 12.5% calorie→12.5% kilocalorie

Answer: We thank the reviewer’s correction and have change to kilocalorie.

REVIEWER COMMENTS

Reviewer #1 (Remarks to the Author):

Dear Authors,

Regarding the answers to my comments, I have some notices:

1. Table S2 as well as Mascot T1683_ATp5a modifications on K125 and K175. The peptide is the same as well as its m/z signal as well as its fragmentation. Mascot attributes this peptide to the same protein one time in mitochondria and one not, with the same number of identified peptides. It seems peculiar to me. Please verify. I think the peptide is the same and the modification is on the same K.

2. Immunoblottings (Figure 3a and b). The authors report a single WB analysis of HSP70 as loading control and they refer all the densitometric analysis to the same HSP70 blot. However, they show immunoblottings for five different proteins (UCP1, PRDM16, PGC1a, DIO2 and CIDEA) and, in particular, UCP1 and DIO2 have the same MW (also CIDEA is really near as MW). Do the authors run two or more immunoblottings for these proteins? Do they have HSP70 blots for different runs? Please in the methods section add that UCP1 and DIO2 antibodies are from rabbit.

3. The proteomics on BT should be performed also after the use of AD-9308 to fully complete the work.

Reviewer #2 (Remarks to the Author):

I carefully went over the manuscript, and acknowledge that the manuscript has improved substantially. But I am still not aligned with the authors correction of energy expenditure by lean body mass. The authors here state that the energy expenditure was corrected according to the recommendation of an international guide for measuring metabolic rates (Matthias H Tschöp, et al. Nature Methods 2012; 9:57-63). But this stated guideline paper does explicitly say that energy expenditure should not be divided by lean body mass. In

fact, I wrote part of that paper myself. Correction of energy expenditure by lean body mass erroneously overcompensates for the lean mass effect, as has been shown subsequently in several other manuscripts (McMurray F et al., PLoS Genet. 2013;9(1):e1003166 and Muller TD et al.

Nat Metab. 2021 Sep;3(9):1134-1136 and Fernandez-Verdejo R et al., Nat Methods 16, 797-799). The authors shall please show a regression analysis with Energy expenditure (kcal/h) vs. body weight (g) or lean mass (g) and perform an ANCOVA, as suggested step-by-step here (Speakman JR et al., Dis Model Mech. 2013 Mar;6(2):293-301). Other than that, the paper has improved nicely and is certainly worth being published in this journal.

Reviewer #3 (Remarks to the Author):

the authors have adequately addressed the comments

Response to Reviewer's comments

Reviewer #1 (Remarks to the Author):

Dear Authors,

Regarding the answers to my comments, I have some notices:

1. Table S2 as well as Mascot T1683_ATp5a modifications on K125 and K175. The peptide is the same as well as its m/z signal as well as its fragmentation. Mascot attributes this peptide to the same protein one time in mitochondria and one not, with the same number of identified peptides. It seems peculiar to me. Please verify. I think the peptide is the same and the modification is on the same K.

Our response: We thank the reviewer's careful correction. Please see the aligned peptide sequence shown below, which is also deposited in the he ProteomeXchange public dataset base (<https://massive.ucsd.edu/ProteoSAFe/static/massive.jsp>). These two aligned peptides matched to the sequence of D3Z6F5_MOUSE (ATP synthase subunit alpha, Atp5a1) and Q03262_MOUSE (ATP synthase F1 subunit alpha, Atp5fa1). *Atp5a1* and *Atp5fa1* are the same gene (gene ID: 498) and their encoded proteins differ only in the first 50 amino acids, which are mitochondrial transit signal peptides.

ALDH2 WT BAT

Protein View: D3Z6F5_MOUSE (D3Z6F5)

Score:2425

Monoisotopic mass (Mr):54675

Calculated pI:8.24

Protein sequence coverage: 54%

1 **MSSILEERIL** **GADTSVDLEE** **TGRVLSIGDG** **IARVHGLRNV** **QAEEMVEFSS**
 51 **GLKGMSLNLE** **PDNVGVVVF** **NDKLIKEGDV** **VKRTGAIVDV** **PVGEELLGRV**
 101 **VDALGNAIDG** **KGPIGSKTRR** **RVGLKAPGII** **PRISVREPMQ** **TGIKAVDSLV**
 151 **PIGRGQRELI** **IGDRQTGKTS** **IAIDTIINQK** **RFNDGTDEKK** **KLYCIYVAIG**
 201 **QKRSTVAQLV** **KRLTDADAMK** **YTIVVSATAS** **DAAPLQYLAP** **YSGCSMGEYF**
 251 **RDNGKHALII** **YDDLKQAVA** **YRQMSLLLR** **PPGREAYPGD** **VFYLHSRLLLE**
 301 **RAAKMNSDFG** **GGSLTALPVI** **ETQAGDVSAY** **IPTNVISITD** **GQIFLETELF**
 351 **YKGIRPAINV** **GLSVSRVGS** **AQTRAMKQVA** **GTMKLELAQY** **REVAFAQFG**
 401 **SDLDAATQQL** **LSRGVRLTEL** **LKQGQYSPMA** **IEEQVAVIYA** **GVRGYLDKLE**
 451 **PSKITKFENA** **FLSHVISQHQ** **SLLGNI** **RSDG** **KISEQSDAKL** **KEIVTNFLAG**
 501 FEP

Query Start – End	Observed	Mr(expt)	Mr(calc)ppm	M	Score	ExpectRank	U	Peptide
52519 169 – 180	491.6194	1471.8363	1471.8497-9.13	0	49	0.0029	1	K.TSIAIDTIINQK.R + HNE (K)

ALDH2 KI BAT

Protein View: ATPA_MOUSE (Q03265)

Score:1320

Monoisotopic mass (Mr):59830

Calculated pI:9.22

Protein sequence coverage: 48%

1 **MLSVRVAAAV** **ARALPRRAGL** **VSKNALGSSF** **VGARNLHASN** **TRLQKTGTAE**
 51 **MSSILEERIL** **GADTSVDLEE** **TGRVLSIGDG** **IARVHGLRNV** **QAEEMVEFSS**
 101 **GLKGMSLNLE** **PDNVGVVVF** **NDKLIKEGDV** **VKRTGAIVDV** **PVGEELLGRV**
 151 **VDALGNAIDG** **KGPIGSKTRR** **RVGLKAPGII** **PRISVREPMQ** **TGIKAVDSLV**
 201 **PIGRGQRELI** **IGDRQTGKTS** **IAIDTIINQK** **RFNDGTDEKK** **KLYCIYVAIG**
 251 **QKRSTVAQLV** **KRLTDADAMK** **YTIVVSATAS** **DAAPLQYLAP** **YSGCSMGEYF**
 301 **RDNGKHALII** **YDDLKQAVA** **YRQMSLLLR** **PPGREAYPGD** **VFYLHSRLLLE**
 351 **RAAKMNSDFG** **GGSLTALPVI** **ETQAGDVSAY** **IPTNVISITD** **GQIFLETELF**
 401 **YKGIRPAINV** **GLSVSRVGS** **AQTRAMKQVA** **GTMKLELAQY** **REVAFAQFG**
 451 **SDLDAATQQL** **LSRGVRLTEL** **LKQGQYSPMA** **IEEQVAVIYA** **GVRGYLDKLE**
 501 **PSKITKFENA** **FLSHVISQHQ** **SLLGNI** **RSDG** **KISEQSDAKL** **KEIVTNFLAG**
 551 FEP

Query Start – End	Observed	Mr(expt)	Mr(calc)ppm	M	Score	ExpectRank	U	Peptide
44893 171 – 182	358.9860	1431.9150	1431.9289-9.75	2	30	0.025	1	U.R.VGLKAPGIIPR.I + HNE (K)
47710 219 – 230	491.6192	1471.8359	1471.8497-9.43	0	38	0.041	1	U.K.TSIAIDTIINQK.R + HNE (K)
47711 219 – 230	491.6192	1471.8359	1471.8497-9.43	0	54	0.00095	1	U.K.TSIAIDTIINQK.R + HNE (K)
47710 219 – 230	491.6192	1471.8359	1471.8497-9.43	0	38	0.041	1	U.K.TSIAIDTIINQK.R + HNE (K)
47711 219 – 230	491.6192	1471.8359	1471.8497-9.43	0	54	0.00095	1	U.K.TSIAIDTIINQK.R + HNE (K)

Therefore, as the reviewer suggested, they should be the same proteins. We have corrected this error in the revised **Fig. 3i&j** and **Supplementary Table S2** as follows and in the revised manuscript. However, the incorporation of *Atp5a1* (originally assigned to wild-type-specific 4-HNE-adducted proteins) to *Atp5fla* (originally assigned to 4-HNE-adducted proteins shared by both *Aldh2* wild-type and knock-in mice) do not change the conclusion because the identified 4-HNE-adducted proteins specific to knock-in mice remain same. Please see the revised **Results** and **Fig. 3i&j** as follows.

“Using liquid-chromatography tandem mass spectrometry (LC MS/MS) analysis, we identified 19 4-HNE-adducted brown adipose tissue mitochondrial proteins in *Aldh2* knock-in mice and 9 4-HNE adducted mitochondrial proteins in wild-type mice, with 8 proteins which are present in both *Aldh2* knock-in and wild-type mice (**Fig. 3i, 3j**).”

Figure 3i

Figure 3j

Gene name	protein name	modified site	biological function
Aldh2 knock-in mice			
Acaa2	3-ketoacyl-CoA thiolase, mitochondrial	K25	fatty acid beta-oxidation
Pcca	Propionyl-CoA carboxylase alpha chain, mitochondrial	K275	propanoyl-CoA degradation
Ndufc2	NADH dehydrogenase [ubiquinone] 1 subunit C2	H8	mitochondrial electron transport
Ndufb4	NADH dehydrogenase [ubiquinone] 1 beta subcomplex subunit 4	H59	mitochondrial electron transport
Sdhb	Succinate dehydrogenase [ubiquinone] iron-sulfur subunit, mitochondrial	H246	mitochondrial electron transport chain ,tricarboxylic acid cycle
Pdha1	Pyruvate dehydrogenase E1 component subunit alpha, somatic form, mitochondrial	H121	Tricarboxylic acid cycle
Aco2	Aconitate hydratase, mitochondrial	K144	Tricarboxylic acid cycle
Hspa8	Heat shock cognate 71 kDa protein (627aa)	K168	ATP binding
Gpd2	Glycerol-3-phosphate dehydrogenase	K652 K668	glycerol-3-phosphate metabolic process
Pcx	Pyruvate carboxylase	H574	pyruvate metabolic process
Myh4	Myosin-4	K1525	Muscle contraction
Aldh2 knock-in & wild-type mice			
Acaa2	3-ketoacyl-CoA thiolase, mitochondrial	K375	fatty acid beta-oxidation
Uqcrb	Cytochrome b-c1 complex subunit 7	H39	mitochondrial electron transport
Uqcrc1	Cytochrome b-c1 complex subunit 1, mitochondrial	K3	mitochondrial electron transport
Uqcrrf1	Cytochrome b-c1 complex subunit Rieske, mitochondrial	K101	mitochondrial electron transport
Immt	MICOS complex subunit MIC60	K296	maintenance of mitochondrial architecture
Immt	MICOS complex subunit MIC60(isoform)	K218	maintenance of mitochondrial architecture
Atp5f1a (Atp5a1)	ATP synthase subunit alpha, mitochondrial	K175 (K125)	mitochondrial electron transport
Hspa8	Heat shock cognate 71 kDa protein (646aa)	K187	MAPK signaling pathway
Aldh2 wild-type mice			
Chchd3	MICOS complex subunit	K24	maintenance of mitochondrial architecture

Table S2. SI table of identified 4-hydroxynonenal (4-HNE)-adducted protein by liquid chromatography tandem mass spectrometry (LC-MS/MS)

Aldh2 knock-in mice							
Uniprot ID	Protein name	Gene name	Mascot score	Peptide Num.	emPAI	Sequence	Adducted residue
Q8BWT1	3-ketoacyl-CoA thiolase, mitochondrial	Acca2	3226	31	29.48	R.TPFGAYGGLL K .D + HNE (K)	K25
Q91ZA3	Propionyl-CoA carboxylase alpha chain, mitochondrial	Pcca	715	15	1.28	R.HIEIQVLGD K HGNALWLNRECSIQR.R +HNE (H)	H275
Q9CQ54	NADH dehydrogenase [ubiquinone] 1 subunit C2	Ndufc2	351	4	2.14	M. M NGRPG H EPLKFLPDEAR.S + HNE (H)	H8
Q9CQC7	NADH dehydrogenase [ubiquinone] 1 beta subcomplex subunit 4	Ndufb4	220	4	2.85	R.VSH I EDPALIR.W + HNE (H)	H59
Q9CQA3	Succinate dehydrogenase [ubiquinone] iron-sulfur subunit, mitochondrial	Sdhb	1366	16	15.7	R.C H TIMNCTQTCPK.G + HNE (H)	H246
P35486	Pyruvate dehydrogenase E1 component	Pdha1	747	18	5.58	R.A H GFTFTR.G + HNE (H)	H121

	subunit alpha, somatic form, mitochondrial						
Q99KI0	Aconitate hydratase, mitochondrial	Aco2	5325	47	16.12	R.AK D INQEVYNFLATAGAK.Y + HNE (K)	K144
Q504P4	Heat shock cognate 71 kDa protein (627aa)	Hspa8	211	6	0.43	R.IINEPTAAAIAYGLD K .K + HNE (K)	K168
A2AQR0	Glycerol-3-phosphate dehydrogenase	Gpd2	3813	40	7.92	R.FH K FDEDEKGFITIVDVQR.V + HNE (K)	K652
						R.FH K FDEDE K GFITIVDVQR.V + HNE (K)	K658
G5E8R3	Pyruvate carboxylase	Pcx	2028	33	2.15	R.DA H QSLLATR.V + HNE (H)	H574
Q5SX39	Myosin-4	Myh4	601	16	0.37	K.NLQQEISDLTEQIAEGG K .H + HNE (K)	K1525
Aldh2 knock-in & wild-type mice							
Uniprot ID	Protein name	Gene name	Mascot score	Peptide Num.	emPAI	Sequence	Adducted residue
Q8BWT1	3-ketoacyl-CoA thiolase, mitochondrial	Acaa2	2275	30	21.75	R.GG K YAVGSACIGGGQGIALLIQNTA.- + HNE (K)	K375
Q9CQB4	Cytochrome b-c1 complex subunit 7	Uqcrb	967	7	13.84	R.DDTL H ETEDVKEAIR.R + HNE (H)	H39

A0A0A6YVZ 0	Cytochrome b-c1 complex subunit 1, mitochondrial	Uqcrc1	133	2	9.36	-.DE K NNGAGYFLEHLAFK.L + HNE (K)	K3
Q9CR68	Cytochrome b-c1 complex subunit Rieske, mitochondrial	Uqcrcf1	466	9	2.49	R.AEVL DST K SSKESSEAR.K + HNE (K)	K101
Q8CAQ8	MICOS complex subunit Mic60	Immt	1829	36	5.49	R.KAVDEAADALL K .A + HNE (K)	K296
E9Q800	MICOS complex subunit Mic60	Immt	1826	35	6.54	R.KAVDEAADALL K .A + HNE (K)	K218
Q03265	ATP synthase subunit alpha, mitochondrial	Atp5f1a (Atp5a1)	1118	22	3.58	R.RVGL K APGIIPR.I + HNE (K)	K175
D3Z6F5	ATP synthase subunit alpha	Atp5a1	1067	22	4.28	R.RVGL K APGIIPR.I + HNE (K)	K125
P63017	Heat shock cognate 71 kDa protein	Hspa8	281	7	0.5	R.IINEPTAAAIAYGLD K .K + HNE (K)	187
Aldh2 wild-type mice							
Uniprot ID	Protein name	Gene name	Mascot score	Peptide Num.	emPAI	Sequence	Adducted residue
D3Z0L4	MICOS complex subunit	Chchd3	555	11	7.88	R.VTFEADENENITVV K .G + HNE (K)	K24

2. Immunoblottings (Figure 3a and b). The authors report a single WB analysis of HSP70 as loading control and they refer all the densitometric analysis to the same HSP70 blot. However, they show immunoblottings for five different proteins (UCP1, PRDM16, PGC1 α , DIO2 and CIDEA) and, in particular, UCP1 and DIO2 have the same MW (also CIDEA is really near as MW). Do the authors run two or more immunoblottings for these proteins? Do they have HSP70 blots for different runs? Please in the methods section add that UCP1 and DIO2 antibodies are from rabbit.

Our response: We thank the reviewer's careful correction. Indeed, the immunoblots experiment were run in separate gels. Therefore, we added the internal controls for each immunoblot and the corresponding histogram as follows (please also see revised Fig. 3a&3b). The uncropped and bright-field images are enclosed in Supplementary Fig.18 folder.

3. The proteomics on BT should be performed also after the use of AD-9308 to fully complete the work.

Our response: We thank the reviewer's comments. We performed LC-MS/MS to identify the 4-HNE-adducted proteins in *Aldh2* wild-type (WT) (n=3:3) and knock-in (KI) mice (n=3:3) receiving or not receiving AD-9308 treatment.

Treatment of AD-9308 reduced the number of 4-NHE-adducted proteins in the BAT from *Aldh2* WT mice from 14 to 4. Among the 10 proteins, 8 proteins were involved in mitochondrial electron transfer and fatty acid beta-oxidation including the Trifunctional enzyme subunit alpha, mitochondrial (*Hadha*), Succinate dehydrogenase [ubiquinone] flavoprotein subunit, mitochondrial (*Sdha*), Cytochrome b-c1 complex subunit 1, mitochondrial (*Uqcrc1*), Long-chain-fatty-acid--CoA ligase (*Acs11*), Trifunctional enzyme subunit beta, mitochondrial (*Hadhb*), 3-ketoacyl-CoA thiolase, mitochondrial (*Acaa2*), Cytochrome b-c1 complex subunit 7 (*Uqcrb*), and Cytochrome c oxidase subunit 6C (*Cox6c*) being eliminated by AD9308. Please see the revised **Results, Supplementary Fig. 15,** and **Supplementary Table S2** as follows.

“Finally, to identify 4-HNE-adducted proteins modified by AD-9308 treatment, we performed LC-MS/MS analyses of the BAT isolated from *Aldh2* WT and KI mice treated or not treated with AD-9308. As shown in **Supplementary Fig.15** and **Table S2**, treatment of AD-9308 reduced the number of 4-NHE-adducted proteins in the BAT from *Aldh2* WT mice from 14 to 4. Among the 10 proteins, 8 proteins are

involved in mitochondrial electron transfer and fatty acid oxidation including the Trifunctional enzyme subunit alpha, mitochondrial (*Hadha*), Succinate dehydrogenase [ubiquinone] flavoprotein subunit, mitochondrial (*Sdha*), Cytochrome b-c1 complex subunit 1, mitochondrial (*Uqcrc1*), Long-chain-fatty-acid--CoA ligase (*Acs11*), Trifunctional enzyme subunit beta, mitochondrial (*Hadhb*), 3-ketoacyl-CoA thiolase, mitochondrial (*Acaa2*), Cytochrome b-c1 complex subunit 7 (*Uqcrb*), and Cytochrome c oxidase subunit 6C (*Cox6c*) being eliminated by AD9308.”

Figure S15. 4-HNE-adducted mitochondrial proteins of the BAT from the *Aldh2* WT mice receiving or not receiving AD9308 identified by liquid-chromatography tandem mass spectrometry (LC-MS/MS) (n=3:3).”

Similarly, treatment of AD-9308 reduced the number of 4-NHE-adducted proteins in the BAT from *Aldh2* KI mice from 18 to 4. Among the 14 proteins, 12 proteins were involved in mitochondrial electron transfer and fatty acid beta-oxidation including the Trifunctional enzyme subunit alpha, mitochondrial (*Hadha*), Succinate dehydrogenase [ubiquinone] flavoprotein subunit, mitochondrial (*Sdha*), Cytochrome b-c1 complex subunit 1, mitochondrial (*Uqcrc1*), Cytochrome b-c1 complex subunit 2, mitochondrial (*Uqcrc2*), Trifunctional enzyme subunit beta, mitochondrial (*Hadhb*), Cytochrome c oxidase subunit NDUFA4 (*Ndufa4*), Mitochondrial carnitine/acylcarnitine carrier protein (*Slc25a20*), Cytochrome b-c1 complex subunit 7 (*Uqcrb*), NADH dehydrogenase [ubiquinone] 1 beta subcomplex subunit 10 (*Ndufb10*), NADH dehydrogenase [ubiquinone] iron-sulfur protein 6, mitochondrial (*Ndufs6*), and Cytochrome c oxidase subunit 6C (*Cox6c*) being eliminated by AD-9308. Please see the revised **Results, Supplementary Fig. 16** and **Supplementary Table S2** as follows.

“Similarly, treatment of AD-9308 reduced the number of 4-NHE-adducted proteins in the BAT from *Aldh2* KI mice from 18 to 4. Among the 14 proteins, 12 proteins are involved in mitochondrial electron transfer and fatty acid beta-oxidation including the Trifunctional enzyme subunit alpha, mitochondrial (*Hadha*), Succinate dehydrogenase [ubiquinone] flavoprotein subunit, mitochondrial (*Sdha*), Cytochrome b-c1 complex subunit 1, mitochondrial (*Uqcrc1*), Cytochrome b-c1 complex subunit 2, mitochondrial (*Uqcrc2*), Trifunctional enzyme subunit beta, mitochondrial (*Hadhb*), Cytochrome c oxidase subunit NDUFA4 (*Ndufa4*), Mitochondrial carnitine/acylcarnitine carrier protein (*Slc25a20*), Cytochrome b-c1 complex subunit 7 (*Uqcrb*), NADH dehydrogenase [ubiquinone] 1 beta subcomplex subunit 10 (*Ndufb10*), NADH dehydrogenase [ubiquinone] iron-sulfur protein 6, mitochondrial (*Ndufs6*), and Cytochrome c

oxidase subunit 6C (*Cox6c*) being eliminated by AD-9308 (**Supplementary Fig. 16** and **Table S2**) These results were consistent with the results comparing 4-HNE-adducted proteins of BAT between *Aldh2* WT and KI mice, showing ALDH2 modulates mitochondrial ETC and FAO function in BAT.

Figure S16. 4-HNE-adducted mitochondrial proteins of the BAT from the *Aldh2* KI mice receiving or not receiving AD9308 identified by liquid-chromatography tandem mass spectrometry (LC-MS/MS) (n=3:3).”

Reviewer #2 (Remarks to the Author):

I carefully went over the manuscript, and acknowledge that the manuscript has improved substantially. But I am still not aligned with the authors correction of energy expenditure by lean body mass. The authors here state that the energy expenditure was corrected according to the recommendation of an international guide for measuring metabolic rates (Matthias H Tschöp, et al. Nature Methods 2012; 9:57-63). But this stated guideline paper does explicitly say that energy expenditure should not be divided by lean body mass. In fact, I wrote part of that paper myself. Correction of energy expenditure by lean body mass erroneously overcompensates for the lean mass effect, as has been shown subsequently in several other manuscripts (McMurray F et al., PLoS Genet. 2013;9(1):e1003166 and Muller TD et al. Nat Metab. 2021 Sep;3(9):1134-1136 and Fernandez-Verdejo R et al., Nat Methods 16, 797-799). The authors shall please show a regression analysis with Energy expenditure (kcal/h) vs. body weight (g) or lean mass (g) and perform an ANCOVA, as suggested step-by-step here (Speakman JR et al., Dis Model Mech. 2013 Mar;6(2):293-301). Other than that, the paper has improved nicely and is certainly worth being published in this journal.

Our response: We thank the reviewer’s constructive comments and the useful reference. All energy expenditure among three genotypes was re-analyzed using the generalized linear model according to

international guidance (reference 60, 61). Energy expenditure (expressed as kcal/hr) of each mouse was regressed on fat and lean mass using the command “glm” implemented in STATA 14.0 and the three genotypes were coded by allelic doses as 0,1, and 2 for wild-type, heterozygous, and homozygous knock-in mice, respectively. We thank the assistant professor Pi-Hua Liu of the Clinical Informatics and Medical Statistics Research Center of Chang Gung University in Taiwan for statistical assistance.

The revised **Methods**, **Fig. 2a&2h**, **Supplementary Fig. 2**, & **3b**, and **acknowledgement** are shown below.

“Statistical analyses

Energy expenditure among three genotypes was analyzed using the generalized linear model according to international guidance^{60, 61}. Energy expenditure (expressed as kcal/hr) was regressed on fat and lean mass using the command “glm” implemented in STATA 14.0 and the three genotypes were coded by allelic doses as 0,1, and 2 for wild-type, heterozygous, and homozygous knock-in mice, respectively^{60,61}.

Figure 2a

Figure 2. (a) Energy expenditure (n=19:27:13) of the *Aldh2* homozygous knock-in (KI), heterozygous knock-in (HE), and wild-type (WT) mice.

Figure 2h

Figure 2. (h) Isoproterenol -induced energy expenditure (n=4:8:4) of *Aldh2* homozygous KI, HE, and WT mice. The arrow head indicate the injection time.

Supplementary Figure S2

Figure S2. Diet-induced increased energy expenditure of *Aldh2* wild-type (WT), heterozygous knock-in (HE), and homozygous knock-in mice (KI). Energy expenditure measured by indirect calorimetry after high-fat high-sucrose feeding for 4 hours after overnight fasting. (n=10:14:8).

Supplementary Figure S3b

Figure S3. (b) Energy expenditure in acute cold tolerance tests of *Aldh2* homozygous knock-in (KI) and heterozygous knock-in (HE) mice compared with wild-type (WT) mice (n=6:6:4)

Acknowledgement

We thank the assistant professor Pi-Hua Liu of the Clinical Informatics and Medical Statistics Research Center of Chang Gung University in Taiwan for statistical assistance.

References

60. Tschöp, M.H., et al. A guide to analysis of mouse energy metabolism. *Nat. Methods.* **9**, 57-63. (2011).
61. Speakman, J.R., Fletcher, Q., & Vaanholt, L. The '39 steps': an algorithm for performing statistical analysis of data on energy intake and expenditure. *Dis Model Mech.* **6**:293-301 (2013)."

Reviewer #3 (Remarks to the Author):

the authors have adequately addressed the comments

Our response: We thank the reviewer's encouragement.

REVIEWERS' COMMENTS

Reviewer #1 (Remarks to the Author):

The authors correctly answered to my requests

Reviewer #2 (Remarks to the Author):

I thank the authors for their corrected data. I am now happy with the results and applaud the authors' eagerness and perseverance. The paper is nice and deserves its home in this journal.

Response to Reviewer's comments

Reviewer #1 (Remarks to the Author):

The authors correctly answered to my requests

Our response: We thank the review's encouragement.

Reviewer #2 (Remarks to the Author):

I thank the authors for their corrected data. I am now happy with the results and applaud the authors' eagerness and perseverance. The paper is nice and deserves its home in this journal.

Our response: We thank the review's encouragement.